# Learning curves for continual learning in neural networks: Self-knowledge transfer and forgetting

**Ryo Karakida & Shotaro Akaho**
National Institute of Advanced Industrial Science and Technology (AIST), Japan
{karakida.ryo,s.akaho}@aist.go.jp

## Abstract

Sequential training from task to task is becoming one of the major objects in deep learning applications such as continual learning and transfer learning. Nevertheless, it remains unclear under what conditions the trained model's performance improves or deteriorates. To deepen our understanding of sequential training, this study provides a theoretical analysis of generalization performance in a solvable case of continual learning. We consider neural networks in the neural tangent kernel (NTK) regime that continually learn target functions from task to task, and investigate the generalization by using an established statistical mechanical analysis of kernel ridge-less regression. We first show characteristic transitions from positive to negative transfer. More similar targets above a specific critical value can achieve positive knowledge transfer for the subsequent task while catastrophic forgetting occurs even with very similar targets. Next, we investigate a variant of continual learning which supposes the *same* target function in multiple tasks. Even for the same target, the trained model shows some transfer and forgetting depending on the sample size of each task. We can guarantee that the generalization error monotonically decreases from task to task for equal sample sizes while unbalanced sample sizes deteriorate the generalization. We respectively refer to these improvement and deterioration as *self*-knowledge transfer and forgetting, and empirically confirm them in realistic training of deep neural networks as well.

## 1 Introduction

As deep learning develops for a single task, it enables us to work on more complicated learning frameworks where the model is sequentially trained on multiple tasks, e.g., transfer learning, curriculum learning, and continual learning. Continual learning deals with the situation in which the learning machine cannot access previous data due to memory constraints or privacy reasons. It has attracted much attention due to the demand on applications, and fundamental understanding and algorithms are being explored (Hadsell et al., 2020). One well-known phenomenon is catastrophic forgetting; when a network is trained between different tasks, naive training cannot maintain performance on the previous task (McCloskey & Cohen, 1989; Kirkpatrick et al., 2017).

It remains unclear in most cases under what conditions a trained model's performance improves or deteriorates in sequential training. Understanding its generalization performance is still limited (Pentina & Lampert, 2014; Bennani et al., 2020; Lee et al., 2021). For single-task training, however, many empirical and theoretical studies have succeeded in characterizing generalization performance in over-parameterized neural networks and given quantitative evaluation on sample-size dependencies, e.g., double descent (Nakkiran et al., 2020). For further development, it will be helpful to extend the analyses on single-task training to sequential training on multiple tasks and give theoretical backing.

To deepen our understanding of sequential training, this study shows a theoretical analysis of its generalization error in the neural tangent kernel (NTK) regime. In more details, we consider the NTK formulation of continual learning proposed by Bennani et al. (2020); Doan et al. (2021). By extending a statistical mechanical analysis of kernel ridge-less regression, we investigate learning curves, i.e., the dependence of generalization error on sample size or number of tasks. The analysis

focuses on the continual learning with explicit task boundaries. The model learns data generated by similar teacher functions, which we call target functions, from one task to another. All input samples are generated from the same distribution in an i.i.d. manner, and each task has output samples (labels) generated by its own target function. Our main contributions are summarized as follows. First, we revealed characteristic transitions from negative to positive transfer depending on the target similarity. More similar targets above a specific critical value can achieve positive knowledge transfer (i.e., better prediction on the subsequent task than training without the first task). Compared to this, backward transfer (i.e., prediction on the previous task) has a large critical value, and subtle dissimilarity between targets causes negative transfer. The error can rapidly increase, which clarifies that catastrophic forgetting is literally catastrophic (Section 4.1).

Second, we considered a variant of continual learning, that is, learning of the *same* target function in multiple tasks. Even for the same target function, the trained model's performance improves or deteriorates in a non-monotonic way. This depends on the sample size of each task; for equal sample sizes, we can guarantee that the generalization error monotonically decreases from task to task (Section 4.2 for two tasks & Section 5 for more tasks). Unbalanced sample sizes, however, deteriorates generalization (Section 4.3). We refer to these improvement and deterioration of generalization as self-knowledge transfer and forgetting, respectively. Finally, we empirically confirmed that self-knowledge transfer and forgetting actually appear in the realistic training of multi-layer perceptron (MLP) and ResNet-18 (Section 5.1). Thus, this study sheds light on fundamental understanding and the universal behavior of sequential training in over-parameterized learning machines.

## 2 RELATED WORK

**Method of analysis.** As an analysis tool, we use the replica method originally developed for statistical mechanics. Statistical mechanical analysis enables us *typical-case* evaluation, that is, the average evaluation over data samples or parameter configurations. It sometimes provides us with novel insight into what the worst-case evaluation has not captured (Abbaras et al., 2020; Spigler et al., 2020). The replica method for kernel methods has been developed in Dietrich et al. (1999), and recently in Bordelon et al. (2020); Canatar et al. (2021). These recent works showed excellent agreement between theory and experiments on NTK regression, which enables us to quantitatively understand sample-size dependencies including implicit spectral bias, double descent, and multiple descent.

**Continual learning.** Continual learning dynamics in the NTK regime was first formulated by Bennani et al. (2020); Doan et al. (2021), though the evaluation of generalization remains unclear. They derived an upper bound of generalization via the Rademacher complexity, but it includes naive summation over single tasks and seems conservative. In contrast, our typical-case evaluation enables us to newly find such rich behaviors as negative/positive transfer and self-knowledge transfer/forgetting. The continual learning in the NTK regime belongs to so-called single-head setting (Farquhar & Gal, 2018), and it allows the model to revisit the previous classes (target functions) in subsequent tasks. This is complementary to earlier studies on incremental learning of new classes and its catastrophic forgetting (Ramasesh et al., 2020; Lee et al., 2021), where each task includes different classes and does not allow the revisit. Note that the basic concept of continual learning is not limited to incremental learning but allows the revisit (McCloskey & Cohen, 1989; Kirkpatrick et al., 2017). Under the limited data acquisition or resources of memory, we often need to train the same model with the same targets (but different samples) from task to task. Therefore, the setting allowing the revisit seems reasonable.

## 3 PRELIMINARIES

### 3.1 NEURAL TANGENT KERNEL REGIME

We summarize conventional settings of the NTK regime (Jacot et al., 2018; Lee et al., 2019). Let us consider a fully connected neural network $f = u_L$ given by

$$u_l = \sigma_w W_l h_{l-1}/\sqrt{M_{l-1}} + \sigma_b b_l, \quad h_l = \phi(u_l) \quad (l = 1, ..., L), \tag{1}$$

where we define weight matrices $W_l \in \mathbb{R}^{M_l \times M_{l-1}}$, bias terms $b_l \in \mathbb{R}^{M_l}$, and their variances $\sigma_w^2$ and $\sigma_b^2$. We set random Gaussian initialization $W_{l,ij}, b_{l,i} \sim \mathcal{N}(0, 1)$ and focus on the mean squared error loss: $\mathcal{L}(\theta) = \sum_{\mu=1}^N \|y^\mu - f(x^\mu)\|^2$, where the training samples $\{x^\mu, y^\mu\}_{\mu=1}^N$ are composed of

inputs $x^\mu \in \mathbb{R}^D$ normalized by $\|x^\mu\|_2 = 1$ and labels $y^\mu \in \mathbb{R}^C$. The set of all parameters is denoted as $\theta$, and the number of training samples is $N$. Assume the infinite-width limit for hidden layers ($M_l \to \infty$), finite sample size and depth. The gradient descent dynamics with a certain learning rate then converges to a global minimum sufficiently close to the random initialization. This is known as the NTK regime, and the trained model is explicitly obtained as

$$f^{(c)}(x') = f_0^{(c)}(x') + \Theta(x', X)\Theta(X)^{-1}(y^{(c)} - f_0^{(c)}(X)) \quad (c = 1, ..., C). \tag{2}$$

We denote the NTK matrix as $\Theta$, arbitrary input samples (including test samples) as $x'$, and the set of training samples as $X$. The indices of $f$ mean 0 for the model at initialization and $c$ for the head of the network. Entries of NTK $\Theta(x', x)$ are defined by $\nabla_\theta f_0(x')\nabla_\theta f_0(x)^\top$. We write $\Theta(X)$ as an abbreviation for $\Theta(X, X)$. The trained network is equivalent to a linearized model around random initialization (Lee et al., 2019), that is, $f^{(c)} = f_0^{(c)} + \nabla_\theta f_0^{(c)}\Delta\theta$ with

$$\Delta\theta = \theta - \theta_0 = \sum_{c=1}^{C} \nabla_\theta f_0^{(c)}(X)^\top \Theta(X)^{-1}(y^{(c)} - f_0^{(c)}(X)). \tag{3}$$

While over-parameterized models have many global minima, the NTK dynamics implicitly select the above $\theta$, which corresponds to the L2 min-norm solution. Usually, we ignore $f_0$ by taking the average over random initialization. The trained model (2) is then equivalent to the kernel ridge-less regression (KRR).

NTK regime also holds in various architectures including ResNets and CNNs (Yang & Littwin, 2021), and the difference only appears in the NTK matrix. Although we focus on the fully connected network in synthetic experiments, the following NTK formulation of sequential training and our theoretical results hold in any architecture under the NTK regime.

**NTK formulation of Continual learning.** We denote the set of training samples in the $n$-th task as $(X_n, y_n)$ ($n = 1, 2, ..., K$), a model trained in a sequential manner from task 1 to task $n$ as $f_n$ and its parameters as $\theta_n$. That is, we train the network initialized at $\theta_{n-1}$ for the $n$-th task and obtain $f_n$. Assume that the number of tasks $K$ is finite. The trained model within the NTK regime is then given as follows (Bennani et al., 2020; Doan et al., 2021):

$$f_n(x') = f_{n-1}(x') + \Theta(x', X_n)\Theta(X_n)^{-1}(y_n - f_{n-1}(X_n)), \tag{4}$$

$$\theta_n - \theta_{n-1} = \nabla_\theta f_0(X_n)^\top \Theta(X_n)^{-1}(y_n - f_{n-1}(X_n)). \tag{5}$$

We omit the index $c$ because each head is updated independently. The model $f_n$ completely fits the $n$-th task, i.e., $y_n = f_n(X_n)$. The main purpose of this study is to analyze the generalization performance of the sequentially trained model (4). At each task, the model has an inductive bias of KRR in the function space and L2 min-norm solution in the parameter space. The next task uses this inductive bias as the initialization of training. The problem is whether this inductive bias helps improve the generalization in the subsequent tasks.

**Remark (independence between different heads).** For a more accurate understanding of the continual learning in the NTK regime, it may be helpful to remark on the heads' independence, which previous studies did not explicitly mention. As in Eq. (2), all heads share the same NTK, and $f^{(c)}$ depends only on the label of its class $y^{(c)}$. While the parameter update (3) includes information of all classes, the $c$-th head can access only the information of the $c$-th class[1]. For example, suppose that the $n$-th task includes all classes except 1, i.e., $\{2, ..., C\}$. Then, the model update on the the class 1 at the $n$-th task, i.e., $f_n^1 - f_{n-1}^1$, becomes

$$\nabla_\theta f_0^{(1)}(x')(\theta_n - \theta_{n-1}) = \nabla_\theta f_0^{(1)}(x') \sum_{c=2} \nabla_\theta f_0^{(c)}(X_n)^\top \Theta(X_n)^{-1}(y_n^{(c)} - f_{n-1}^{(c)}) = 0$$

because $\nabla_\theta f_0^{(c)}\nabla_\theta f_0^{(c')\top} = 0$ ($c \neq c'$) in the infinite-width limit (Jacot et al., 2018; Yang, 2019). Thus, we can deal with each head independently and analyze the generalization by setting $C = 1$ without loss of generality. This indicates that in the NTK regime, interaction between different heads do not cause knowledge transfer and forgetting. One may wonder if there are any non-trivial knowledge transfer and forgetting in such a regime. Contrary to such intuition, we reveal that when the subsequent task revisits previous classes (targets), the generalization shows interesting increase and decrease.

---

[1]We use the term "class", although the regression problem is assumed in NTK theory. Usually, NTK studies solve the classification problem by regression with a target $y^{(c)} = \{0, 1\}$.

### 3.2 LEARNING CURVE ON SINGLE-TASK TRAINING

To evaluate the generalization performance of (4), we extend the following theory to our sequential training. Bordelon et al. (2020) obtained an analytical expression of the generalization for NTK regression on a single task (2) as follows. Assume that training samples are generated in an i.i.d. manner ( $x^\mu \sim p(x)$ ) and that labels are generated from a square integrable target function $\bar{f}$:

$$\bar{f}(x) = \sum_{i=0}^{\infty} \bar{w}_i \psi_i(x), \quad y^\mu = \bar{f}(x^\mu) + \varepsilon^\mu \qquad (\mu = 1, ..., N), \qquad (6)$$

where $\bar{w}_i$ are constant coefficients and $\varepsilon$ represents Gaussian noise with $\langle \varepsilon^\mu \varepsilon^\nu \rangle = \delta_{\mu\nu}\sigma^2$. We define $\psi_i(x) := \sqrt{\eta_i}\phi_i(x)$ with basis functions $\phi_i(x)$ given by Mercer's decomposition:

$$\int dx' p(x') \Theta(x, x') \phi_i(x') = \eta_i \phi_i(x) \qquad (i = 0, 1, \ldots, \infty). \qquad (7)$$

Here, $\eta_i$ denotes NTK's eigenvalue and we assume the finite trance of NTK $\sum_i \eta_i < \infty$. We set $\eta_0 = 0$ in the main text to avoid complicated notations. We can numerically compute eigenvalues by using the Gauss-Gegenbauer quadrature. Generalization error is expressed by $E_1 := \left\langle \int dx p(x) \left( \bar{f}(x) - f^*(x) \right)^2 \right\rangle_{\mathcal{D}}$ where $f^*$ is a trained model and $\langle \cdots \rangle_{\mathcal{D}}$ is the average over training samples. Bordelon et al. (2020) derived a typical-case evaluation of the generalization error by using the replica method: for a sufficiently large $N$, we have asymptotically

$$E_1 = \frac{1}{1 - \gamma} \sum_{i=0}^{\infty} \eta_i \bar{w}_i^2 \left( \frac{\kappa}{\kappa + N\eta_i} \right)^2 + \frac{\gamma}{1 - \gamma} \sigma^2. \qquad (8)$$

Although the replica method takes a large sample size $N$, Bordelon et al. (2020); Canatar et al. (2021) reported that the analytical expression (8) coincides well with empirical results even for small $N$. The constants $\kappa$ and $\gamma$ are defined as follows and characterize the increase and decrease of $E_1$:

$$1 = \sum_{i=0}^{\infty} \frac{\eta_i}{\kappa + N\eta_i}, \quad \gamma = \sum_{i=0}^{\infty} \frac{N\eta_i^2}{(\kappa + N\eta_i)^2}. \qquad (9)$$

The $\kappa$ is a positive solution of the first equation and obtained by numerical computation. The $\gamma$ satisfies $0 < \gamma < 1$ by definition. For $N = \alpha D^l$ ($\alpha > 0, l \in \mathbb{N}, D \gg 1$), we can analytically solve it and obtain more detailed evaluation. For example, a positive $\kappa$ decreases to zero as the sample size increases and $E_1$ also decreases for $\sigma^2 = 0$. For $\sigma^2 > 0$, the generalization error shows multiple descent depending on the decay of eigenvalue spectra. Since multiple descent is not a main topic of this paper, we briefly summarize it in Section A.5 of the Supplementary Materials.

## 4 LEARNING CURVES BETWEEN TWO TASKS

In this section, we analyze the NTK formulation of continual learning (4) between two tasks ($K = 2$). One can also regard this setting as transfer learning. We sequentially train the model from task A to task B, and each one has a target function defined by

$$\bar{f}_A(x) = \sum_i \bar{w}_{A,i} \psi_i(x), \quad \bar{f}_B(x) = \sum_i \bar{w}_{B,i} \psi_i(x), \quad [\bar{w}_{A,i}, \bar{w}_{B,i}] \sim \mathcal{N}(0, \eta_i \begin{bmatrix} 1 & \rho \\ \rho & 1 \end{bmatrix}). \qquad (10)$$

The target functions are dependent on each other and belong to the reproducing kernel Hilbert space (RKHS). By denoting the RKHS by $\mathcal{H}$, one can interpret the target similarity $\rho$ as the inner product $\langle \bar{f}_A, \bar{f}_B \rangle_{\mathcal{H}} / (\|\bar{f}_A\|_{\mathcal{H}} \|\bar{f}_B\|_{\mathcal{H}}) = \rho$. These targets have dual representation such as $\bar{f}(x) = \sum_i \alpha_i \Theta(x_i', x)$ with i.i.d. Gaussian variables $\alpha_i$ (Bordelon et al., 2020), as summarized in Section E.1. We generate training samples by $y_A^\mu = \bar{f}_A(x_A^\mu) + \varepsilon_A^\mu$ ($\mu = 1, ..., N_A$) and $y_B^\mu = \bar{f}_B(x_B^\mu) + \varepsilon_B^\mu$ ($\mu = 1, ..., N_B$), although we focus on the noise-less case ($\sigma = 0$) in this section. Input samples $x_A^\mu$ and $x_B^\mu$ are i.i.d. and generated by the same distribution $p(x)$. We can measure the generalization error in two ways: generalization error on subsequent task B and that on previous task A:

$$E_{A \to B}(\rho) = \left\langle \int dx p(x) (\bar{f}_B(x) - f_{A \to B}(x))^2 \right\rangle, \qquad (11)$$

$$E_{A \to B}^{back}(\rho) = \left\langle \int dx p(x) (\bar{f}_A(x) - f_{A \to B}(x))^2 \right\rangle, \qquad (12)$$

where we write $f_2$ as $f_{A \to B}$ to emphasize the sequential training from A to B. The notation $E_{A \to B}^{back}(\rho)$ is referred to as backward transfer (Lopez-Paz & Ranzato, 2017). Large negative backward transfer is known as catastrophic forgetting. We take the average $\langle \cdots \rangle$ over training samples of two tasks $\{\mathcal{D}_A, \mathcal{D}_B\}$, and target coefficients $\bar{w}$. In fact, we can set $\bar{w}$ as constants, and it is unnecessary to take the average. To avoid complicated notation, we take the average in the main text. We then obtain the following result.

**Theorem 1.** *Using the replica method under sufficiently large $N_A$ and $N_B$, for $\sigma = 0$, we have*

$$E_{A \to B}(\rho) = \sum_i \left[ 2(1 - \rho)(1 - q_{A,i}) + \frac{q_{A,i}^2}{1 - \gamma_A} \right] E_{B,i}, \tag{13}$$

$$E_{A \to B}^{back}(\rho) = \sum_i \left[ 2(1 - \rho)(1 + F_{\gamma_B}(q_{A,i}, q_{B,i}))\eta_i^2 + \frac{q_{A,i}^2}{1 - \gamma_A} E_{B,i} \right], \tag{14}$$

*where we define $q_{A,i} = \kappa_A/(\kappa_A + N_A \eta_i)$, $q_{B,i} = \kappa_B/(\kappa_B + N_B \eta_i)$, $E_{B,i} = q_{B,i}^2 \eta_i^2/(1 - \gamma_B)$ and $F_{\gamma_B}(a, b) = b(a - 2) + b^2(1 - a)/(1 - \gamma_B)$.*

The constants $\kappa_A$ and $\gamma_A$ ($\kappa_B$ and $\gamma_B$, respectively) are given by setting $N = N_A(N_B)$ in (9). The detailed derivation is given in Section A. Technically speaking, we use the following lemma:

**Lemma 2.** *Denote the trained model (2) on single task A as $f_A = \sum_i w_{A,i}^* \psi_i$, and define the following cost function: $E = \left\langle \sum_i \phi_i (w_{A,i}^* - u_i)^2 \right\rangle_{\mathcal{D}_A}$ for arbitrary constants $\phi_i$ and $u_i$. Using the replica method under a sufficiently large $N_A$, we have*

$$E = \sum_i \left[ (\bar{w}_{A,i} - u_i)^2 - 2\bar{w}_{A,i}(\bar{w}_{A,i} - u_i)q_{A,i} + \left\{ \bar{w}_{A,i}^2 + \frac{\eta_i N_A}{\kappa_A^2}(E_A + \sigma^2) \right\} q_{A,i}^2 \right] \phi_i,$$

*where $E_A$ denotes generalization error $E_1$ on single task A.*

For example, we can see that $E_A$ is a special case of $E$ with $\phi_i = \eta_i$ and $u = \bar{w}_A$, and that $E_{A \to B}(\rho)$ is reduced to $\phi_i = q_{B,i}^2 \eta_i^2/(1 - \gamma_B)$ and $u = \bar{w}_B$ after certain calculation.

**Spectral Bias.** The generalization errors obtained in Theorem 1 are given by the summation over spectral modes like the single-task case. The $E_{B,i}$ corresponds to the $i$-th mode of generalization error (8) on single task B. The study on the single task (Bordelon et al., 2020) revealed that as the sample size increases, the modes of large eigenvalues decreases first. This is known as *spectral bias* and clarifies the inductive bias of the NTK regression. Put the eigenvalues in descending order, i.e., $\lambda_i \geq \lambda_{i+1}$. When $D$ is sufficiently large and $N_B = \alpha D^l$ ($\alpha > 0$, $l \in \mathbb{N}$), the error of each mode is asymptotically given by $E_{B,i} = 0$ ($i < l$) and $\eta_i^2$ ($i > l$). We see that the spectral bias also holds in $E_{A \to B}$ because it is a weighted sum over $E_{B,i}$. In contrast, $E_{A \to B}^{back}$ includes a constant term $2(1 - \rho)\eta_i^2$. This constant term causes catastrophic forgetting, as we show later.

### 4.1 TRANSITION FROM NEGATIVE TO POSITIVE TRANSFER

We now look at more detailed behaviors of generalization errors obtained in Theorem 1. We first discuss the role of the target similarity $\rho$ for improving generalization. Figure 1(a) shows the comparison of the generalization between single-task training and sequential training. Solid lines show theory, and markers show experimental results of trained neural networks in the NTK regime. We trained the model (1) with ReLU activation, $L = 3$, and $M_l = 4,000$ by using the gradient descent over 50 trials. More detailed settings of our experiments are summarized in Section E. Because we set $N_A = N_B = 100$, we have $E_A = E_B$. The point is that both $E_{A \to B}(\rho)$ and $E_{A \to B}^{back}(\rho)$ are lower than $E_A(= E_B)$ for large $\rho$. This means that the sequential training degrades generalization if the targets are dissimilar, that is, negative transfer. In particular, $E_{A \to B}^{back}(\rho)$ rapidly deteriorates for the dissimilarity of targets. Note that both $E_{A \to B}$ and $E_{A \to B}^{back}$ are linear functions of $\rho$. Figure 1(a) indicates that the latter has a large slope. We can gain quantitative insight into the critical value of $\rho$ for the negative transfer as follows.

**Knowledge transfer.** The following asymptotic equation gives us the critical value for $E_{A \to B}$:

$$E_{A \to B}(\rho)/E_B \sim 2(1 - \rho) \qquad for \quad N_A \gg N_B. \tag{15}$$

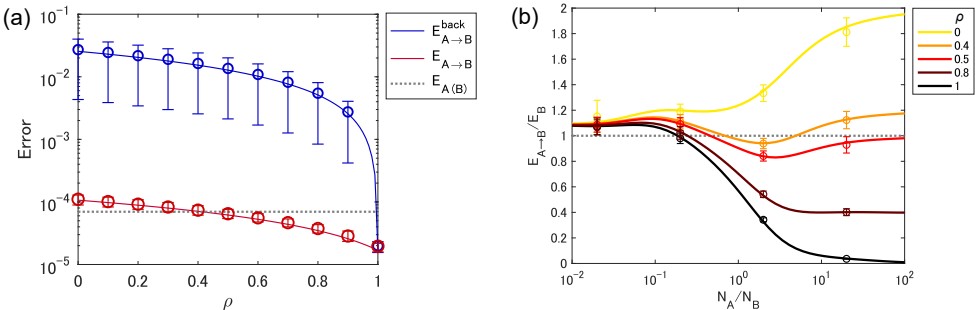

Figure 1: (a) Transitions from positive to negative transfer caused by target similarity $\rho$. We set $N_A = N_B$. (b) Learning curves show negative transfer $(E_{A \to B}/E_B)$ in a highly non-linear way depending on unbalanced sample sizes. We changed $N_A$ and set $N_B = 10^3$.

A straightforward algebra leads to this (Section A.3). In the context of transfer learning, it is reasonable that the target domain has a limited sample size compared to the first task. For $\rho > 1/2$, we have $E_{A \to B} < E_B$, that is, previous task A contributes to improving the generalization performance on subsequent task B (i.e., positive transfer). For $\rho < 1/2$, however, negative transfer appears.

The following sufficient condition for the negative transfer is also noteworthy. By evaluating $E_{A \to B} > E_B$, we can prove that for any $N_A$ and $N_B$, the negative transfer always appears for

$$\rho < \rho^* := \sqrt{\gamma_A}/(1 + \sqrt{\gamma_A}). \tag{16}$$

This is just a sufficient condition and may be loose. For example, we asymptotically have the critical target similarity $\rho = 1/2 > \rho^*$ for $N_A \gg N_B$. Nevertheless, this sufficient condition is attractive in the sense that it clarifies the unavoidable negative transfer for the small target similarity.

**Backward transfer.** The $E_{A \to B}^{back}(0)$ includes the constant term $\sum_i \eta_i^2$ independent of sample sizes. Note that $q_A$ and $q_B$ decrease to zero for large sample sizes (Bordelon et al., 2020), and we have $E_{A \to B}^{back}(\rho) \sim \sum_i \eta_i^2(1 - \rho)$. In contrast, $E_A$ converges to zero for a large $N_A$. Therefore, the intersection between $E_{A \to B}^{back}(\rho)$ and $E_A$ reach $\rho = 1$ as $N_A$ and $N_B$ increase. This means that when we have a sufficient number of training samples, negative backward transfer $(E_{A \to B}^{back}(\rho) > E_A)$ occurs even for very similar targets. Figure 1(a) confirms this result, and Figure 6 in Section A.6 shows more detailed learning curves of backward transfer. Catastrophic forgetting seems literally "catastrophic" in the sense that the backward transfer rapidly deteriorates by the subtle target dissimilarity.

## 4.2 SELF-KNOWLEDGE TRANSFER

We have shown that target similarity is a key factor for knowledge transfer. We reveal that the sample size is another key factor. To clarify the role of sample sizes, we focus on the same target function ($\rho = 1$) in the following analysis. We refer to the knowledge transfer in this case as *self-knowledge transfer* to emphasize the network learning the same function by the same head. As is the same in $\rho < 1$, the knowledge obtained in the previous task is transferred as the network's initialization for the subsequent training and determines the eventual performance.

**Positive transfer by equal sample sizes.** We find that positive transfer is guaranteed under equal sample sizes, that is, $N_A = N_B$. To characterize the advantage of the sequential training, we compare it with a model average: $(f_A + f_B)/2$, where $f_A$ ($f_B$) means the model obtained by a single-task training on A (B). Note that since the model is a linear function of the parameter in the NTK regime, this average is equivalent to that of the trained parameters: $(\theta_A + \theta_B)/2$. After straightforward calculation in Section D, the generalization error of the model average is given by

$$E_{ave} = (1 - \gamma_B/2) \, E_B. \tag{17}$$

Sequential training and model average include information of both tasks A and B; thus, it is interesting to compare it with $E_{A \to B}$. We find

**Proposition 3.** *For $\rho = 1$ and any $N_A = N_B$,*

$$E_{A \to B}(1) < E_{ave} < E_A = E_B. \tag{18}$$

The derivation is given in Section D. This proposition clarifies the superiority of sequential training over single-task training and even the average. The first task contributes to the improvement on the second task; thus, we have positive transfer.

**Negative transfer by unbalanced sample sizes.** While equal sample size leads to positive transfer, the following unbalanced sample sizes cause the negative transfer of self-knowledge:

$$E_{A \to B}(1)/E_B \sim 1/(1 - \gamma_A) \quad for \quad N_B \gg N_A. \tag{19}$$

The derivation is based on Jensen's inequality (Section A.3). While $E_{A \to B}$ and $E_B$ asymptotically decrease to zero for the large $N_B$, their ratio remains finite. Because $0 < \gamma_A < 1$, $E_{A \to B}(1) > E_B$. It indicates that the small sample size of task A leads to a bad initialization of subsequent training and makes the training on task B hard to find a better solution.

Figure 1(b) summarizes the learning curves which depend on sample sizes in a highly non-linear way. Solid lines show theory, and markers show the results of NTK regression over 100 trials. The figure shows excellent agreement between theory and experiments. Although we have complicated transitions from positive to negative transfer, our theoretical analyses capture the basic characteristics of the learning curves. For self-knowledge transfer ($\rho = 1$), we can achieve positive transfer at $N_A/N_B = 1$, as shown in Proposition 3, and for large $N_A/N_B$, as shown in (15). In contrast, negative transfer appears for small $N_A/N_B$, as shown in (19). For $\rho < 1$, large $N_A/N_B$ produces positive transfer for $\rho < 1/2$, as shown in (15).

If we set a relatively large $\sigma > 0$, the learning curve may become much more complicated due to multiple descent. Figure 5 in Section A.5 confirms the case in which multiple descent appears in $E_{A \to B}$. The curve shape is generally characterized by the interaction among target similarity, self-knowledge transfer depending on sample size, and multiple descent caused by the noise.

### 4.3 SELF-KNOWLEDGE FORGETTING

We have shown in Section 4.1 that backward transfer is likely to cause catastrophic forgetting for $\rho < 1$. We show that even for $\rho = 1$, sequential training causes forgetting. That is, the training on task B degrades generalization even though both tasks A and B learn the same target.

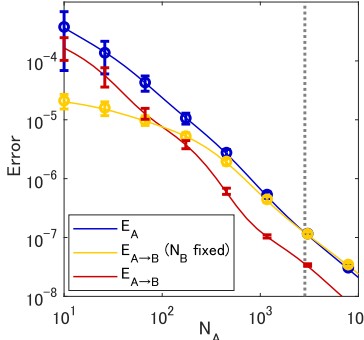

We have $E_{A \to B}^{back}(1) = E_{A \to B}(1)$ by definition, and Proposition 3 tells us that $E_{A \to B} < E_A$. Therefore, no forgetting appears for equal sample sizes. In contrast, we have

$$E_{A \to B}(1)/E_A \sim 1/(1 - \gamma_B) \quad for \quad N_A \gg N_B. \tag{20}$$

One can obtain this asymptotic equation in the same manner as (19) since $E_{A \to B}(1)$ is a symmetric function in terms of indices A and B. Combining (20) with (15), we have $E_A < E_{A \to B}(1) \ll E_B$. Sequential training is better than using only task B, but training only on the first task is the best. One can say that the model forgets the target despite learning the same one. We call this self-knowledge forgetting. In the

Figure 2: Self-knowledge forgetting: unbalanced sample sizes degrade generalization.

context of continual learning, many studies have investigated the catastrophic forgetting caused by different heads (i.e., in the situation of incremental learning). Our results suggest that even the sequential training on *the same task and the same head* shows such forgetting. Intuitively speaking, the self-knowledge forgetting is caused by the limited sample size of task B. Note that we have $E_{A \to B}(1) = \sum_i q_{A,i}^2 E_{B,i}/(1 - \gamma_A)$. The generalization error of single-task training on task B ($E_{B,i}$) takes a large value for a small $N_B$ and this causes the deterioration of $E_{A \to B}$ as well. Figure 2 confirms the self-knowledge forgetting in NTK regression. We set $N_A = N_B$ as the red line and $N_B = 100$ as the yellow line. The dashed line shows the point where forgetting appears.

## 5 LEARNING CURVES OF MANY TASKS

We can generalize the sequential training between two tasks to that of more tasks. For simplicity, let us focus on the self-knowledge transfer ($\rho = 1$) and equal sample sizes. Applying Lemma 2 recursively from task to task, we can evaluate generalization defined by $E_n = \left\langle \int dx p(x)(\bar{f}(x) - f_n(x))^2 \right\rangle_{\mathcal{D}}$.

**Theorem 4.** *Assume that (i) $(X_n, y_n)$ $(n = 1, ..., K)$ are given by the same distribution $X_n \sim P(X)$ and target $y_n = \sum_i \bar{w}_i \psi(X_n) + \varepsilon_n$. (ii) sample sizes are equal: $N_n = N$. For $n = 1, ..., K$,*

$$E_{n+1} = \frac{1}{(1-\gamma)^2} \tilde{q}^\top \mathcal{Q}^{n-1} \tilde{q} + R_{n+1}\sigma^2, \quad \mathcal{Q} := \text{diag}\left(q^2\right) + \frac{N}{(1-\gamma)\kappa^2} \tilde{q}\tilde{q}^\top, \quad (21)$$

*where $q_i = \kappa/(\kappa + \eta_i N)$, $\tilde{q}_i = \eta_i q_i^2$ and $diag(q^2)$ denotes a diagonal matrix whose entries are $q_i^2$. The noise term $R_n$ is a positive constant. In the noise-less case ($\sigma = 0$), the learning curve shows monotonic decrease: $E_{n+1} \leq E_n$. If all eigenvalues are positive, we have*

$$E_{n+1} < E_n \qquad (n = 1, 2, ...). \quad (22)$$

See Section B for details of derivation. The learning curve (i.e., generalization error to the number of tasks) monotonically decreases for the noise-less case. the monotonic decrease comes from $\lambda_{max}(\mathcal{Q}) < 1$. This result means that the self-knowledge is transferred and accumulated from task to task and contributes in improving generalization. It also ensures that no self-knowledge forgetting appears. We can also show that $R_n$ converges to a positive constant term for a large $n$ and the contribution of noise remains as a constant.

**KRR-like expression.** The main purpose of this work was to address the generalization of the continually trained model $f_n$. As a side remark, we show another expression of $f_n$:

$$[\Theta\left(x', X_1\right) \cdots \Theta\left(x', X_n\right)] \begin{bmatrix} \Theta\left(X_1\right) & O & \cdots & O \\ \Theta\left(X_2, X_1\right) & \Theta\left(X_2\right) & & \vdots \\ \vdots & & \ddots & O \\ \Theta\left(X_n, X_1\right) & \cdots & & \Theta\left(X_n\right) \end{bmatrix}^{-1} \begin{bmatrix} y_1 \\ \vdots \\ \vdots \\ y_n \end{bmatrix}. \quad (23)$$

This easily comes from comparing the update (4) with a formula for the inversion of triangular block matrices (Section C). One can see this expression as an approximation of KRR, where the upper triangular block of NTK is set to zero. Usual modification of KRR is diagonal, e.g., L2 regularization and block approximation, and it seems that the generalization error of this type of model has never been explored. Our result revealed that this model provides non-trivial sample size dependencies such as self-knowledge transfer and forgetting.

## 5.1 EXPERIMENTS

Although Theorem 4 guarantees the monotonic learning curve for equal sizes, unbalanced sample sizes should cause a non-monotonic learning curve. We empirically confirmed this below.

**Synthetic data.** First, we empirically confirm $E_n$ on synthetic data (10). Figure 3(a1) confirms the claim of Theorem 4 that the generalization error monotonically decreases for $\sigma = 0$ as the task number increases. Dashed lines are theoretical values calculated using the theorem, and points with error bars were numerically obtained by $f_n$ (4) over 100 trials. For $\sigma > 0$, the decrease was suppressed. We set $\sigma^2 = \{0, 10^{-5}, 10^{-4}, 10^{-3}\}$ and $N_i = 100$. Figure 3(a2) shows self-knowledge forgetting. When the first task has a large sample size, the generalization error by the second task can increase for small subsequent sample sizes $N_i$. For smaller $N_i$, there was a tendency for the error to keep increasing and taking higher errors than that of the first task during several tasks. In practice, one may face a situation where the model is initialized by the first task training on many samples and then trained in a continual learning manner under a memory constraint. The figure suggests that if the number of subsequent tasks is limited, we need only the training on the first task. If we have a sufficiently large number of tasks, generalization eventually improves.

**MLP on MNIST / ResNet-18 on CIFAR-10.** We mainly focus on the theoretical analysis in the NTK regime, but it will be interesting to investigate whether our results also hold in more practical settings of deep learning. We trained MLP (fully-connected neural networks with 4 hidden layers) and ResNet-18 with stochastic gradient descent (SGD) and cross-entropy loss. We set the number of epochs sufficient for the training error to converge to zero for each task. We confirmed that they show qualitatively similar results as in the NTK regime. We randomly divided the dataset into tasks without overlap of training samples. Figures 3(b1,c1) show the monotonic decrease for an equal sample size and that the noise suppressed the decrease. We set $N_i = 500$ and generated the noise by the label corruption with a corruption probability $\{0, 0.2, ..., 0.8\}$ (Zhang et al., 2017). The vertical axis means the error, i.e., $1 - $ (Test accuracy [%])/100. Figures 3(b2,c2) show that unbalanced sample sizes caused the non-monotonic learning curve, similar to NTK regression.

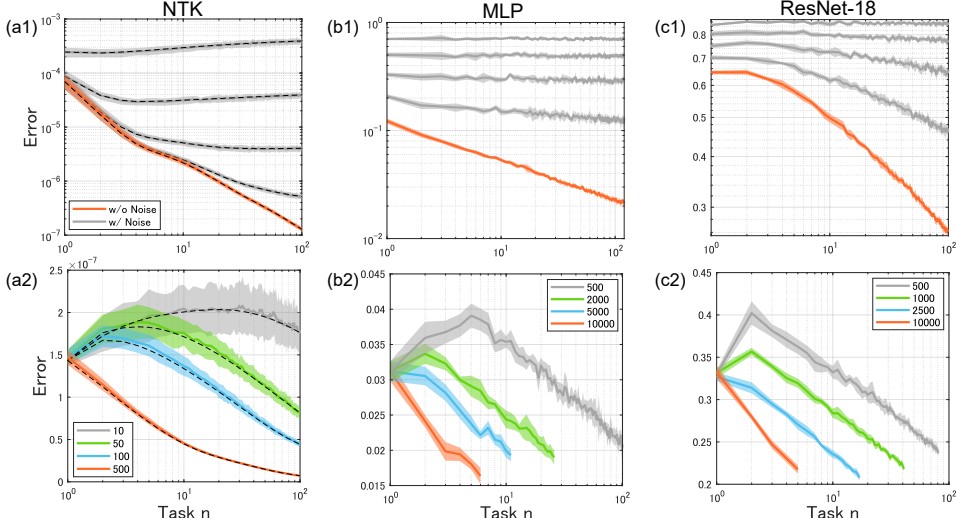

Figure 3: (a1)-(c1) Learning curves for equal sample sizes. For noise-less case, they monotonically decreased (orange lines). For noisy case, decrease was suppressed (grey lines; we plotted learning curves for several $\sigma^2$ and those with larger test errors correspond to larger $\sigma^2$). We trained MLP on MNIST and ResNet-18 on CIFAR-10. (a2)-(c2) Learning curves for unbalanced sample sizes were non-monotonic ($N_1 = 4,000$ for NTK regression, $N_1 = 10^4$ for SGD training of MLP and ResNet-18). Numbers in the legend mean $N_n \quad (n \geq 2)$.

## 6 DISCUSSION

We provided novel quantitative insight into knowledge transfer and forgetting in the sequential training of neural networks. Even in the NTK regime, where the model is simple and linearized, learning curves show rich and non-monotonic behaviors depending on both target similarity and sample size. In particular, learning on the same target shows successful self-knowledge transfer or undesirable forgetting depending on the balance of sample sizes. These results indicate that the performance of the sequentially trained model is more complicated than we thought, but we can still find some universal laws behind it.

There are other research directions to be explored. While we focused on reporting novel phenomena on transfer and forgetting, it is also important to develop algorithms to achieve better performance. To mitigate catastrophic forgetting, previous studies proposed several strategies such as regularization, parameter isolation, and experience replay (Mirzadeh et al., 2020). Evaluating such strategies with theoretical backing would be helpful for further development of continual learning. For example, orthogonal projection methods modify gradient directions (Doan et al., 2021), and replay methods allow the reuse of the previous samples. We conjecture that these could be analyzed by extending our calculations in a relatively straightforward way. It would also be interesting to investigate richer but complicated situations required in practice, such as streaming of non-i.i.d. data and distribution shift (Aljundi et al., 2019). The current work and other theories in continual learning or transfer learning basically assume the same input distribution between different tasks (Lee et al., 2021; Tripuraneni et al., 2020). Extending these to the case with an input distribution shift will be essential for some applications including domain incremental learning. Our analysis may also be extended to topics different from sequential training. For example, self-distillation uses trained model's outputs for the subsequent training and plays an interesting role of regularization (Mobahi et al., 2020).

While our study provides universal results, which do not depend on specific eigenvalue spectra or architectures, it is interesting to investigate individual cases. Studies on NTK eigenvalues have made remarkable progress, covering shallow and deep ReLU neural networks (Geifman et al., 2020; Chen & Xu, 2020), skip connections (Belfer et al., 2021), and CNNs (Favero et al., 2021). We expect that our analysis and findings will serve as a foundation for further understanding and development on theory and experiments of sequential training.

## REPRODUCIBILITY STATEMENT

The main contributions of this work are theoretical claims, and we clearly explained their assumptions and settings in the main text. Complete proofs of the claims are given in the Supplementary Materials. For experimental results of training deep neural networks, we used only already-known models and algorithms implemented in PyTorch. All of the detailed settings, including learning procedures and hyperparameters, are clearly explained in Section E.

## ACKNOWLEDGMENTS

This work was funded by JST ACT-X Grant Number JPMJAX190A and JSPS KAKENHI Grant Number 19K20366.

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

# Supplementary Materials

## A  SEQUENTIAL TRAINING BETWEEN TWO TASKS

### A.1  NOTATIONS

The kernel ridge(-less) regression is given by

$$f^* = \arg\min_{f \in \mathcal{H}} \frac{1}{2\lambda} \sum_{\mu=1}^{N} (f(x^\mu) - y^\mu)^2 + \frac{1}{2} \langle f, f \rangle_{\mathcal{H}} \tag{S.1}$$

where $N$ is the sample size and $\mathcal{H}$ is the reproducing kernel Hilbert space (RKHS) induced by the neural tangent kernel $\Theta$. We assume that the input samples $x^\mu$ are generated from a probabilistic distribution $p(x)$ and that the NTK has finite trace, i.e., $\int dx p(x) \Theta(x, x) < \infty$. Mercer's decomposition of $\Theta$ is expressed by

$$\int dx' p(x') \Theta(x, x') \phi_i(x') = \eta_i \phi_i(x) \qquad (i = 0, 1, \dots, \infty). \tag{S.2}$$

In other words, we have $\Theta(x', x) = \sum_{i=0}^{\infty} \eta_i \phi_i(x') \phi_i(x)$. Note that a function belonging to RKHS is given by

$$f(x) = \sum_{i=0}^{\infty} a_i \phi_i(x) \tag{S.3}$$

with $\|f\|_{\mathcal{H}}^2 = \sum_{i=0}^{\infty} a_i^2 / \eta_i < \infty$. Then, using the orthonormal bases $\{\phi_i\}_{i=0}^{\infty}$, the solution of the regression is given by

$$f^*(x) = \sum_i w_i^* \psi_i(x), \ w^* = \left( \Psi\Psi^\top + \lambda I \right)^{-1} \Psi y \tag{S.4}$$

where $\psi_i(x) := \sqrt{\eta_i} \phi_i(x)$. Each column of $\Psi$ is given by $\phi(x^\mu)$ $(\mu = 1, ..., N)$. We focus on the NTK regime and take the ridge-less limit $(\lambda \to 0)$.

The sequentially trained model (4) between tasks A and B is written as

$$f_{A \to B}(x) - f_A(x) = \Theta(x, X_B) \Theta(X_B)^{-1} (y_B - f_A(X_B)), \tag{S.5}$$

$$f_A(x) = \Theta(x, X_A) \Theta(X_A)^{-1} y_A. \tag{S.6}$$

The model $f_A$ is trained on single task A and represented by

$$f_A(x) = w_A^{*\top} \psi(x) \tag{S.7}$$

with

$$w_A^* = \lim_{\lambda \to 0} \mathrm{argmin}_{w_A} H_A(w_A), \tag{S.8}$$

$$H_A(w_A) := \frac{1}{2\lambda} \sum_{\mu=1}^{N_A} \left( w_A^\top \psi(x_A^\mu) - y_A^\mu \right)^2 + \frac{1}{2} \|w_A\|_2^2. \tag{S.9}$$

Eq. (S.9 ) is the objective function equivalent to (S.1 ) because $\langle f_A, f_A \rangle_{\mathcal{H}} = \|w_A\|_2^2$. Similarly, we can represent $f_{A \to B}$ by the series expansion with the bases of RKHS. Note that the right-hand side of (S.5 ) is equivalent to the kernel ridge-less regression on input samples $X_B$ and labels $y_B - f_A(B)$. We have

$$f_{A \to B}(x) = (w_A^* + w_B^*)^\top \psi(x) \tag{S.10}$$

with

$$w_B^* = \lim_{\lambda \to 0} \mathrm{argmin}_{w_B} H(w_B, w_A^*), \tag{S.11}$$

$$H(w_B, w_A^*) := \frac{1}{2\lambda} \sum_{\mu=1}^{N_B} \left( w_B^\top \psi(x_B^\mu) - (y_B^\mu - w_A^{*\top} \psi(x_B^\mu)) \right)^2 + \frac{1}{2} \|w_B\|_2^2. \tag{S.12}$$

## A.2 Proof of Theorem 1.

### A.2.1 Knowledge transfer $E_{A \to B}$

First, we take the average over training samples of task B conditioned by task A, that is,

$$E_{A \to B|A} := \left\langle \int dx p(x)(\bar{f}_B(x) - (w_A^* + w_B^*)^\top \psi(x))^2 \right\rangle_{\mathcal{D}_B} \tag{S.13}$$

$$= \left\langle \int dx p(x)((w_B^* - (\bar{w}_B - w_A^*))^\top \psi(x))^2 \right\rangle_{\mathcal{D}_B} \tag{S.14}$$

where the set of task B's training samples is denoted by $\mathcal{D}_B$. We have

$$E_{A \to B} = \langle E_{A \to B|A} \rangle_{\mathcal{D}_A}. \tag{S.15}$$

Note that the objective function $H(w_B, w_A^*)$ is transformed to

$$H(w_B, w_A^*) = \frac{1}{2\lambda} \sum_{\mu=1}^{N_B} \left( (w_B - (\bar{w}_B - w_A^*))^\top \psi(x_B^{(\mu)}) \right)^2 + \frac{1}{2} \|w_B\|_2^2. \tag{S.16}$$

Comparing (S.14) and (S.16), one can see that the average over task B conditioned by task A is equivalent to single-task training with the target function $(\bar{w}_B - w_A^*)^\top \psi(x)$. Therefore, by using (8), we immediately obtain

$$E_{A \to B|A} = \frac{1}{1 - \gamma_B} \sum_i \eta_i (\bar{w}_{B,i} - w_{A,i}^*)^2 \left( \frac{\kappa_B}{\kappa_B + N_B \eta_i} \right)^2 + \frac{\gamma_B}{1 - \gamma_B} \sigma^2 \tag{S.17}$$

$$=: \sum_i \phi_i (\bar{w}_{B,i} - w_{A,i}^*)^2 + \frac{\gamma_B}{1 - \gamma_B} \sigma^2. \tag{S.18}$$

Next, we take the average of (S.18) over task A. Only $\bar{w}_A^*$ depends on the task A and is determined by the single-task training (S.8). This corresponds to Lemma 2 with $u = \bar{w}_B$ and $\phi_i = \eta_i q_{B,i}^2/(1 - \gamma_B)$. We obtain

$$E_{A \to B} = \sum_i \Bigg[ (\bar{w}_{A,i} - \bar{w}_{B,i})^2 - 2\bar{w}_{A,i}(\bar{w}_{A,i} - \bar{w}_{B,i})q_{A,i}$$

$$+ \left( \bar{w}_{A,i} + \frac{1}{1 - \gamma_A} \frac{\eta_i N_A}{\kappa_A^2} \sum_j \eta_j \bar{w}_{A,j}^2 q_{A,j}^2 \right) q_{A,i}^2 \Bigg] \frac{\eta_i q_{B,i}^2}{1 - \gamma_B}$$

$$+ \sigma^2 \left( \frac{1}{1 - \gamma_A} \frac{1}{1 - \gamma_B} \frac{N_A}{\kappa_A^2} \sum_i \eta_i^2 q_{A,i}^2 q_{B,i}^2 + \frac{\gamma_B}{1 - \gamma_B} \right). \tag{S.19}$$

Although this is a general result that holds for any $\bar{w}_A$ and $\bar{w}_B$, it is a bit complicated and seems not easy to give an intuitive explanation. Let us take the average over

$$[\bar{w}_{A,i}, \bar{w}_{B,i}] \sim \mathcal{N}(0, \eta_i \begin{bmatrix} 1 & \rho \\ \rho & 1 \end{bmatrix}). \tag{S.20}$$

The generalization error is then simplified to

$$E_{A \to B}(\rho) = \langle E_{A \to B} \rangle_w \tag{S.21}$$

$$= \sum_i \left[ 2(1 - \rho)(1 - q_{A,i}) + \frac{q_{A,i}^2}{1 - \gamma_A} \right] E_{B,i}$$

$$+ \sigma^2 \left( \frac{1}{1 - \gamma_A} \frac{1}{1 - \gamma_B} \frac{N_A}{\kappa_A^2} \sum_i \eta_i^2 q_{A,i}^2 q_{B,i}^2 + \frac{\gamma_B}{1 - \gamma_B} \right), \tag{S.22}$$

where we used $\langle w_{A,i}^2 \rangle_w = \langle w_{B,i}^2 \rangle_w = \eta_i$ and $\langle w_{A,i}, w_{B,i} \rangle_w = \rho$. $\qquad \square$

### A.2.2 BACKWARD TRANSFER

Backward transfer is measured by the prediction on the previous task A:

$$E_{A \to B}^{back} := \left\langle \int dx p(x) (\bar{f}_A(x) - f_{A \to B}(x))^2 \right\rangle \tag{S.23}$$

$$= \left\langle \int dx p(x) ((w_B^* - (\bar{w}_A - w_A^*))^\top \psi(x))^2 \right\rangle. \tag{S.24}$$

First, we take the average over task B,

$$E_{A \to B|A}^{back} = \left\langle \int dx p(x) ((w^* - (\bar{w}_A - w_A^*))^\top \psi(x))^2 \right\rangle_{\mathcal{D}_B}. \tag{S.25}$$

This corresponds to Lemma 2 with the replacement $\phi_i \leftarrow \eta_i$ and $u \leftarrow \bar{w}_A - w_A^*$. Recall that the objective function of NTK regression was given by (S.16). The target $\bar{w}_A$ in Lemma 2 is replaced as $\bar{w}_A \leftarrow \bar{w}_B - w_A^*$. We then have

$$E_{A \to B|A}^{back} = \sum_i \Bigg[ (\bar{w}_{B,i} - \bar{w}_{A,i})^2 - 2(\bar{w}_{B,i} - w_{A,i}^*)(\bar{w}_{B,i} - \bar{w}_{A,i}) q_{B,i} $$
$$+ \left\{ (\bar{w}_{B,i} - w_{A,i}^*)^2 + \frac{1}{1 - \gamma_B} \frac{\eta_i N_B}{\kappa_B^2} (\sum_{j=0} \eta_j (\bar{w}_{B,j} - w_{A,j}^*)^2 q_{B,j}^2 + \sigma^2) \right\} q_{B,i}^2 \Bigg] \eta_i \tag{S.26}$$

$$= \gamma_B \sum_i \eta_i (\bar{w}_{B,i} - \bar{w}_{A,i})^2 + \sum_i \frac{\eta_i q_{B,i}^2}{1 - \gamma_B} \left( w_{A,i} - (\bar{w}_{B,i} - \frac{1 - \gamma_B}{q_{B,i}} (\bar{w}_{B,i} - \bar{w}_{A,i})) \right)^2 $$
$$+ \frac{\gamma_B}{1 - \gamma_B} \sigma^2. \tag{S.27}$$

Next, we take the average over $\mathcal{D}_A$. Since the first and third terms of (S.27) are independent of $\mathcal{D}_A$, we need to evaluate only the second term. The second term corresponds to Lemma 2 with $\phi_i = \eta_i q_{B,i}^2/(1 - \gamma_B)$ and $u_i = \bar{w}_{B,i} - (1 - \gamma_B)/q_{B,i}(\bar{w}_{B,i} - \bar{w}_{A,i})$. We obtain

$$E_{A \to B}^{back} = \left\langle E_{A \to B|A}^{back} \right\rangle_{\mathcal{D}_A}$$

$$= \gamma_B \sum_i \eta_i (\bar{w}_{B,i} - \bar{w}_{A,i})^2 + \sum_i \Bigg[ (\bar{w}_{A,i} - \bar{w}_{B,i})^2 (q_{B,i} - (1 - \gamma_B))^2 $$
$$- 2\bar{w}_{A,i}(\bar{w}_{A,i} - \bar{w}_{B,i})(q_{B,i} - (1 - \gamma_B)) q_{A,i} q_{B,i}$$
$$+ \left\{ \bar{w}_{A,i}^2 + \frac{1}{1 - \gamma_A} \frac{\eta_i N_A}{\kappa_A^2} (\sum_j \eta_j w_{A,j}^2 q_{A,j}^2 + \sigma^2) \right\} q_{B,i}^2 q_{B,i}^2 \Bigg] \frac{\eta_i}{1 - \gamma_B}$$
$$+ \sigma^2 \left( \frac{1}{1 - \gamma_A} \frac{1}{1 - \gamma_B} \frac{N_A}{\kappa_A^2} \sum_i \eta_i^2 q_{A,i}^2 q_{B,i}^2 + \frac{\gamma_B}{1 - \gamma_B} \right). \tag{S.28}$$

Finally, by taking the average over (S.20 ), we have

$$
\begin{aligned}
E_{A \to B}^{back}(\rho) &= \langle E_{A \to B}^{back} \rangle_w \\
&= \sum_i \left[ 2\gamma_B \eta_i^2 (1-\rho) + 2\left(q_{B,i} - (1-\gamma_B)\right)^2 \frac{\eta_i^2}{1-\gamma_B}(1-\rho) \right. \\
&\quad \left. - 2\,\eta_i^2 (q_{B,i} - 1 + \gamma_B)\frac{q_{B,i}q_{A,i}}{1-\gamma_B}(1-\rho) + \frac{1}{1-\gamma_A}\frac{1}{1-\gamma_B}\eta_i^2 q_{A,i}^2 q_{B,i}^2 \right] \\
&\quad + \sigma^2 \left( \frac{1}{1-\gamma_A}\frac{1}{1-\gamma_B}\frac{N_A}{\kappa_A^2}\sum_i \eta_i^2 q_{A,i}^2 q_{B,i}^2 + \frac{\gamma_B}{1-\gamma_B} \right) \\
&= 2(1-\rho)\sum_i \eta_i^2 \left[ 1 + q_{B,i}(q_{A,i} - 2) + \frac{q_{B,i}^2(1-q_{A,i})}{1-\gamma_B} \right] + \frac{1}{1-\gamma_A}\frac{1}{1-\gamma_B} \\
&\quad \times \sum_i \eta_i^2 q_{A,i}^2 q_{B,i}^2 + \sigma^2 \left( \frac{1}{1-\gamma_A}\frac{1}{1-\gamma_B}\frac{N_A}{\kappa_A^2}\sum_i \eta_i^2 q_{A,i}^2 q_{B,i}^2 + \frac{\gamma_B}{1-\gamma_B} \right). \quad \text{(S.29)}
\end{aligned}
$$

$\square$

### A.2.3  LEMMA 2

**Lemma 2.** *Suppose training on single task A, the target of which is given by $\bar{f} = \sum_i \bar{w}_{A,i}\psi_i$, and denote the trained model (2) as $f^* = \sum_i w_{A,i}^*\psi_i$. Define the following cost function:*

$$
E = \left\langle \sum_i \phi_i (w_{A,i}^* - u_i)^2 \right\rangle_{\mathcal{D}_A} \tag{S.30}
$$

*for arbitrary constants $\phi_i$ and $u_i$. Using the replica method under a sufficiently large $N_A$, we have*

$$
E = \sum_{i=0} \left[ (\bar{w}_{A,i} - u_i)^2 - 2\bar{w}_{A,i}(\bar{w}_{A,i} - u_i)q_{A,i} \right. \\
\left. + \left\{ \bar{w}_{A,i}^2 + \frac{1}{1-\gamma_A}\frac{\eta_i N_A}{\kappa_A^2}\left(\sum_{j=0}\eta_j \bar{w}_{A,j}^2 q_{A,j}^2 + \sigma^2\right) \right\} q_{A,i}^2 \right]\phi_i. \tag{S.31}
$$

*Proof.* The process of derivation is similar to Canatar et al. (2021), but we have additional constants $\phi_i$ and $u_i$. They cause several differences in the detailed form of equations.

Define

$$
Z[J] = \int dw_A \exp(-\beta H_A(w_A) + J\frac{\beta N}{2}E(w_A)) \tag{S.32}
$$

where $E(w_A) = \left\langle \sum_i \phi_i (w_{A,i} - u_i)^2 \right\rangle_{\mathcal{D}_A}$. We omit the index A in the following. We have

$$
E = \lim_{\beta \to \infty} \frac{2}{\beta N}\frac{\partial}{\partial J}\langle \log Z[J]\rangle_{\mathcal{D}} \Big|_{J=0}. \tag{S.33}
$$

To evaluate $\langle \log Z \rangle$, we use the replica method (a.k.a. replica trick):

$$
\langle \log Z \rangle_{\mathcal{D}} = \lim_{n \to 0}\frac{1}{n}\left( \langle Z^n \rangle_{\mathcal{D}} - 1 \right) \tag{S.34}
$$

The point of the replica method is that we first calculate $Z^n$ for $n \in \mathbb{N}$ then take the limit of $n$ to zero by treating it as a real number. In addition, we calculate the average $\langle Z^n \rangle_{\mathcal{D}}$ under a replica symmetric ansätz and a Gaussian approximation by following the calculation procedure of the previous works (Dietrich et al., 1999; Bordelon et al., 2020; Canatar et al., 2021).

We have

$$\langle Z^n \rangle_{\mathcal{D}} = \int dW_n \exp\left( -\frac{\beta}{2} \sum_{a=1}^{n} \|w^a\|^2 + \frac{\beta J N}{2} \sum_{a=1}^{n} (w^a - u)^\top \Phi (w^a - u) \right) \langle Q \rangle_{\{x^\mu, \varepsilon^\mu\}}^{N_A},$$
(S.35)

$$Q := \exp\left( -\frac{\beta}{2\lambda} \sum_{a=1}^{n} \left( (w_A^a - \bar{w}_A)^\top \psi(x^\mu) - \varepsilon^\mu \right)^2 \right),$$
(S.36)

where we define $dW_n = \prod_{a=1}^{n} dw^a$ and $\Phi$ is a diagonal matrix whose diagonal entries given by $\phi_i$. We take the shift of $w^a \to w^a + \bar{w}$. Then,

$$\langle Z^n \rangle_{\mathcal{D}} = \int dW_n \underbrace{\exp(-\frac{n\beta}{2} \|\bar{w}\|^2 + \frac{n\beta J N}{2} (\bar{w} - u)^\top \Phi (\bar{w} - u))}_{=: \bar{Z}(J)}$$

$$\cdot \exp(\sum_a -\frac{\beta}{2} \|w^a\|^2 + \frac{\beta J N}{2} w^{a\top} \Phi w^a - \beta k^\top w^a) \langle Q \rangle_{\{x^\mu, \varepsilon^\mu\}}^{N},$$
(S.37)

$$Q = \exp(-\frac{\beta}{2\lambda} \sum_{a=1}^{n} (w^{a\top} \psi(x^\mu) - \varepsilon^\mu)^2),$$
(S.38)

$$\bar{k} := \bar{w} - J N \Phi (\bar{w} - u).$$
(S.39)

First, let us calculate $\langle Q \rangle_{\{x^\mu, \varepsilon^\mu\}}$. This term is exactly the same as appeared in the previous works (Bordelon et al., 2020; Canatar et al., 2021). To describe notations, we overview their derivation. Define $q^a = w^{a\top} \psi(x) + \varepsilon$, which are called order parameters. We approximate the probability distribution of $q^a$ by a multivariate Gaussian:

$$P(\{q^a\}) = \frac{1}{\sqrt{(2\pi)^n \det(C)}} \exp\left( -\frac{1}{2} \sum_{a,b} (q^a - \mu^a) [C^{-1}]_{ab} (q^b - \mu^b) \right),$$
(S.40)

with

$$\mu^a := \langle q^a \rangle_{\{x, \varepsilon\}} = \left\langle w^{a\top} \psi(x) \right\rangle_x + \langle \varepsilon \rangle_\varepsilon = \sqrt{\eta_0} w_0^a,$$
(S.41)

$$C^{ab} := \langle q^a q^b \rangle_{\{x, \varepsilon\}} = w^{a\top} \langle \psi(x)\psi(x)^\top \rangle_x w_b + \langle \varepsilon^a \varepsilon^b \rangle_\varepsilon = w^{a\top} \Lambda w^a + \Sigma^{ab},$$
(S.42)

where $\Sigma^{ab} = \sigma \delta_{ab}$, $\langle \cdots \rangle_x$ denotes the average over $p(x)$ and $\Lambda$ is a diagonal matrix whose entries are $\eta_i$. We have $\sqrt{\eta_0} w_0^a$ in (S.41) since $\eta_0$ corresponds to the constant shift $\phi_0(x) = 1$. Training samples are i.i.d. and we omitted the index $\mu$. We then have

$$\langle Q \rangle_{\{x, \varepsilon\}} = \exp\left( -\frac{1}{2} \log \det \left( I + \frac{\beta}{\lambda} C \right) - \frac{\beta}{2\lambda} \mu^\top \left( I + \frac{\beta}{\lambda} C \right)^{-1} \mu \right),$$
(S.43)

where $I$ denotes the identity matrix. Define conjugate variables $\{\hat{\mu}^a, \hat{C}^{ab}\}$ by the following identity:

$$1 = \mathcal{C} \int \left( \prod_{a \geq b} d\mu^a d\hat{\mu}^a dC^{ab} d\hat{C}^{ab} \right)$$

$$\times \exp\left[ -N \sum_a \hat{\mu}^a \left( \mu^a - w_{A,0}^a \sqrt{\eta_0} \right) - N \sum_{a \geq b} \hat{C}^{ab} \left( C^{ab} - w^{a\top} \Lambda w^b - \Sigma^{ab} \right) \right],$$
(S.44)

where $\mathcal{C}$ denotes an uninteresting constant and we took the conjugate variables on imaginary axes.

Next, we perform the integral over $W_n$. Eq. (S.37) becomes

$$\langle Z^n \rangle_{\mathcal{D}}$$
$$= \mathcal{C} \bar{Z}(J) \int dW_n \prod_{a \geq b} d\Omega^{ab} \exp(-N(\sum_a \hat{\mu}^a \mu^a + \sum_{a \geq b} \hat{C}^{ab} (C^{ab} - \Sigma^{ab}))) \exp[-NG - G_S]$$
(S.45)

where $d\Omega^{ab} = d\mu^a d\hat{\mu}^a dC^{ab} d\hat{C}^{ab}$ and define

$$G = \frac{1}{2}\log\det\left(I + \frac{\beta}{\lambda}C\right) + \frac{\beta}{2\lambda}\mu^\top\left(I + \frac{\beta}{\lambda}C\right)^{-1}\mu, \tag{S.46}$$

$$\exp(-G_S) = \exp\left(\sum_a\left(-\frac{\beta}{2}\|w^a\|^2 + \frac{\beta J N}{2}w^{a\top}\Phi w^a - \beta k^\top w^a\right)\right)$$

$$\times \exp\left[N\sum_a\hat{\mu}^a w_{A,0}^a\sqrt{\eta_0} + N\sum_{a\geq b}\hat{C}^{ab}w^{a\top}\Lambda w^b\right]. \tag{S.47}$$

We can represent $\int dW_n\exp(-G_S)$ by

$$\int dW_n\exp(-G_S) = \prod_i\int dx_i\exp\left(-\frac{\beta}{2}x_i^\top\hat{Q}_i x_i - \beta b_i^\top x_i\right) \tag{S.48}$$

where $x_i \in \mathbb{R}^n$ denotes a vector $[w_i^1, ..., w_i^a, ..., w_i^n]^\top$ and $i$ is the index of kernel's eigenvalue mode $(i = 0, 1, ...)$. We defined

$$\hat{Q}_i = (1 - \phi_i J N)I_n - \frac{\eta_i N}{\beta}(\tilde{C} + \text{diag}(\tilde{C})), \tag{S.49}$$

$$b_i = \bar{k}_i 1_n \quad (i \geq 1), \tag{S.50}$$

$$b_0 = \bar{k}_0 1_n - \frac{N\sqrt{\eta_0}}{\beta}\hat{\mu}, \tag{S.51}$$

where $I_n$ is an $n \times n$ identity matrix and $1_n$ is an $n$-dimensional vector whose all entries are 1. The $G_S$ term includes $\phi$ and $c$ that are specific to our study. When $\phi = \eta$ and $u = \bar{w}$, it is reduced to previous works (Bordelon et al., 2020; Canatar et al., 2021). Taking the integral over $\{x_i\}$, we have

$$\int dW_n\exp(-G_S) = \mathcal{C}\prod_{i=0}\exp(\frac{\beta}{2}b_i^\top\hat{Q}_i^{-1}b_i)/\sqrt{\det\hat{Q}_i}. \tag{S.52}$$

That is,

$$G_S = \frac{1}{2}\sum_{i=0}\log\det\hat{Q}_i - \frac{\beta}{2}\sum_{i=0}b_i^\top\hat{Q}_i^{-1}b_i. \tag{S.53}$$

REPLICA SYMMETRY AND SADDLE-POINT METHOD

Next, we carry out the integral (S.47) by the saddle-point method. Assume the replica symmetry:

$$\mu = \mu^a, \quad r = C^{aa}, \quad c = C^{a\neq b}, \tag{S.54}$$

$$\hat{\mu} = \hat{\mu}^a, \quad \hat{r} = \hat{C}^{aa}, \quad \hat{c} = \hat{C}^{a\neq b}. \tag{S.55}$$

The following three terms are the same as in the previous works:

$$\det\left(I + \frac{\beta}{\lambda}C\right) = \left[1 + \frac{\beta}{\lambda}(r - c)\right]^n\left(1 + n\frac{\beta c}{\lambda + \beta(r - c)}\right), \tag{S.56}$$

$$\left(I + \frac{\beta}{\lambda}C\right)^{-1} = \frac{1}{1 + \frac{\beta}{\lambda}(r - c)}\left(I - \frac{\beta c}{\lambda + \beta(r - c) + n\beta c}11^\top\right), \tag{S.57}$$

$$\hat{\mu}^\top\mu + \sum_{a\geq b}\hat{C}^{ab}(C^{ab} - \Sigma^{ab}) = n\left(\hat{\mu}\mu + \hat{r}(r - \sigma^2) - \frac{1}{2}\hat{q}(q - \sigma^2)\right), \tag{S.58}$$

where $11^\top$ denotes a matrix whose all entries are 1. Regarding the leading term of order $n$ $(n \to 0)$,

$$\log\det\left(I + \frac{\beta}{\lambda}C\right) = n\log\left(1 + \frac{\beta}{\lambda}(r - c)\right) + n\frac{\beta C}{\lambda + \beta(r - c)}, \tag{S.59}$$

$$\mu^\top\left(I + \frac{\beta}{\lambda}C\right)^{-1}\mu = \frac{n\mu^2}{1 + \frac{\beta}{\lambda}(r - c)}. \tag{S.60}$$

Furthermore. we have

$$\hat{Q}_i = (1 - \phi_i JN - \frac{\eta_i N}{\beta}(2\hat{r} - \hat{c}))I_n - \frac{\eta_i N\hat{c}}{\beta}1_n 1_n^\top. \tag{S.61}$$

After straightforward algebra, the leading terms become

$$\log \det \hat{Q}_i = n \ln g_i - n\frac{\eta_i N\hat{c}}{g_i\beta}, \tag{S.62}$$

$$1_n^\top \hat{Q}_i^{-1} 1_n = \frac{n}{g_i}, \tag{S.63}$$

with

$$g_i := 1 - \phi_i JN - \frac{\eta_i N}{\beta}(2\hat{r} - \hat{c}). \tag{S.64}$$

Substituting these leading terms into (S.45 ), we obtain $\langle Z^n \rangle_\mathcal{D} = \mathcal{C} \int d\theta \exp(-nNS(\theta))$ with

$$\begin{aligned}
S(\theta) &= -\frac{\beta J}{2}(\bar{w} - u)^\top \Phi(\bar{w} - u) \\
&= \left(\hat{\mu}\mu + \hat{r}(r - \sigma^2) - \frac{1}{2}\hat{c}(c - \sigma^2)\right) + \frac{1}{2}\left(\log\beta\,(r - c) + \frac{c + \mu^2}{r - c}\right) \\
&\quad + \frac{1}{2N}\sum_{i=0}(\ln g_i - (\frac{\eta_i N\hat{c}}{\beta} + \beta\bar{k}_i^2)/g_i) - \frac{1}{2g_0}(-2\bar{k}_0\hat{\mu}\sqrt{\eta_0} + N\eta_0\hat{\mu}^2/\beta),
\end{aligned} \tag{S.65}$$

where $\theta$ denotes a set of variables $\{r, \hat{r}, c, \hat{c}, \mu, \hat{\mu}\}$. Here, we take $N \gg 1$ and use the saddle-point method. We calculate saddle-point equations $\partial S(\theta^*)/\partial\theta = 0$ and obtain

$$\hat{\mu} = -\frac{\mu}{r - c}, \tag{S.66}$$

$$\hat{r} = \frac{1}{2}\left(\frac{c + \mu^2}{(r - c)^2} - \frac{1}{r - c}\right), \tag{S.67}$$

$$\hat{c} = \frac{c + \mu^2}{(r - c)^2}, \tag{S.68}$$

$$\mu = \frac{N\eta_0\hat{\mu}}{g_0\beta} - \frac{\bar{k}_0}{g_0}\sqrt{\eta_0}, \tag{S.69}$$

$$r = \sum_{i=0}(\frac{\eta_i N\hat{c}}{\beta} + \beta\bar{k}_i^2)\frac{\eta_i}{g_i^2\beta} + \frac{\eta_0 N}{g_0^2\beta}(-2\bar{k}_0\hat{\mu}\sqrt{\eta_0} + N\eta_0\hat{\mu}^2\beta) + \sigma^2, \tag{S.70}$$

$$c = \sum_{i=0}(\frac{\eta_i N\hat{c}}{\beta} + \beta\bar{k}_i^2)\frac{\eta_i}{g_i^2\beta} - \frac{1}{\beta}\sum_{i=0}\frac{\eta_i}{g_i} + \frac{\eta_0 N}{g_0^2\beta}(-2\bar{k}_0\hat{\mu}\sqrt{\eta_0} + N\eta_0\hat{\mu}^2\beta) + \sigma^2. \tag{S.71}$$

From (S.67 ) and (S.68 ),

$$r - c = -\frac{1}{2\hat{r} - \hat{c}}. \tag{S.72}$$

From (S.70 ) and (S.71 ), we also have

$$\beta(r - c) = \sum_{i=0}\frac{\eta_i}{g_i} =: \kappa. \tag{S.73}$$

Substituting (S.64 ) and (S.72 ) into (S.73 ), We obtain the implicit function for $\kappa$:

$$1 = \sum_{i=0}\frac{\eta_i}{W(\phi_i)}, \quad W(\phi_i) := \kappa(1 - \phi_i JN) + \eta_i N. \tag{S.74}$$

Using $\kappa$ and $W$, the saddle-point equations become

$$\hat{\mu} = -\frac{\beta\mu}{\kappa}, \tag{S.75}$$

$$\hat{r} = \frac{1}{2}\left(\frac{c+\mu^2}{\kappa^2}\beta^2 - \frac{\beta}{\kappa}\right), \tag{S.76}$$

$$\hat{c} = \frac{\beta^2}{\kappa^2}(c+\mu^2), \tag{S.77}$$

$$\mu = -\frac{\sqrt{\eta_0}\bar{k}_0\kappa}{N\eta_0 + W(\phi_0)}, \tag{S.78}$$

$$r = c + \frac{\kappa}{\beta}. \tag{S.79}$$

Substituting back these quantities into (S.65 ), we obtain

$$S(\theta^*) = -\frac{\beta J}{2}(\bar{w}-u)^\top \Phi(\bar{w}-u) + \frac{1}{2}\ln\kappa + \frac{1}{2N}\sum_{i=0}\ln g_i - \frac{\beta}{2N}\sum_{i=0}\frac{\bar{k}_i^2}{g_i}$$

$$+ \frac{\beta\eta_0\kappa}{2W(\phi_0)}\frac{\bar{k}_0^2}{N\eta_0 + W(\phi_0)} + \frac{\beta}{2\kappa}\sigma^2 + const. \tag{S.80}$$

Recall $\langle Z^n\rangle_{\mathcal{D}} \approx \exp(-nNS(\theta^*))$. We have

$$\langle \log Z\rangle_{\mathcal{D}} = -NS(\theta^*). \tag{S.81}$$

GENERALIZATION ERROR

We evaluate

$$E = \lim_{\beta\to\infty}\frac{2}{\beta N}\frac{\partial}{\partial J}\langle\log Z\rangle_{\mathcal{D}}\bigg|_{J=0} = -\lim_{\beta\to\infty}\frac{2}{\beta}\frac{\partial}{\partial J}S(\theta^*)\bigg|_{J=0}. \tag{S.82}$$

Note that $W, g, \kappa$ and $\bar{k}$ depend on $J$. At the point of $J = 0$,

$$\frac{\partial W(\phi_i)}{\partial J} = -\phi_i N\kappa + \frac{\partial\kappa}{\partial J}, \tag{S.83}$$

$$\frac{\partial g_i}{\partial J} = -(\phi_i + \frac{1}{\kappa^2}\frac{\partial\kappa}{\partial J}\eta_i)N, \tag{S.84}$$

$$\frac{\partial\kappa}{\partial J} = \frac{\kappa^2 N}{1-\gamma}\sum_{i=0}\frac{\eta_i\phi_i}{(\kappa+\eta_i N)^2}, \tag{S.85}$$

$$\frac{\partial\bar{k}_i}{\partial J} = -N\phi_i(\bar{w}_i - u_i). \tag{S.86}$$

Substituting (S.83 -S.86 ) into (S.82 ), we obtain

$$E =$$

$$\sum_{i=0}\left[(\bar{w}_i - u_i)^2 - 2\bar{w}_i(\bar{w}_i - u_i)q_i + \left\{\bar{w}_i^2 + \frac{1}{1-\gamma}\frac{\eta_i N}{\kappa^2}\left(\sum_{j=0}\eta_j\bar{w}_j^2 q_j^2 + \sigma^2\right)\right\}q_i^2\right]\phi_i + \xi_0, \tag{S.87}$$

where $\xi_0$ is

$$\xi_0 = \eta_0\bar{w}_0^2\left[\left(\kappa\frac{N\eta_0 + 2W_0}{W_0^2(N\eta_0 + W_0)^2} - \frac{1}{W_0(N\eta_0 + W_0)}\right)\left(\frac{N}{1-\gamma}\sum_i\eta_i q_i^2\phi_i\right)\right.$$

$$\left. - \phi_0 N\kappa^2\frac{N\eta_0 + 2W_0}{W_0^2(N\eta_0 + W_0)^2}\right] + 2\eta_0\frac{\phi_0\kappa N\bar{w}_0(\bar{w}_0 - u_0)}{W_0(N\eta_0 + W_0)} \tag{S.88}$$

with $W_0 = \kappa + N\eta_0$. When $\eta_0 = 0$, we have $\xi_0 = 0$. Note that the additional term $\xi_0$ is not specific to our case but also appeared in the previous work (Canatar et al., 2021). We have $\xi_0 = \mathcal{O}(1/\eta_0)$ for a large $\eta_0$ and $\xi_0$ is often negligibly small.

$\square$

### A.3 Equations for understanding negative transfer

#### A.3.1 Derivation of equation 15

First, let us evaluate the term including $q_{A,i} = \kappa_A/(\kappa_A + N_A \eta_i)$. Note that $\kappa_A$ is a monotonically decreasing function of $N_A$ because

$$\frac{\partial \kappa_A}{\partial N_A} = -\left(\sum_i \frac{\eta_i}{(\kappa_A + N_A \eta_i)^2}\right)^{-1} \sum_i \frac{\eta_i^2}{(\kappa_A + N_A \eta_i)^2} < 0, \tag{S.89}$$

which comes from the implicit function theorem. Let us write $\kappa_A$ at a finite $N_A = c$ as $\kappa_A(c)$. We then have

$$q_{A,i} \leq \frac{\kappa_A(c)}{N_A \eta_i}, \tag{S.90}$$

for $N_A > c$. Next, we evaluate

$$E_{A \to B}(\rho) = 2(1-\rho)E_B - 2(1-\rho)\sum_i q_{A,i}E_{B,i} + \sum_i \frac{q_{A,i}^2}{1-\gamma_A}E_{B,i}. \tag{S.91}$$

The second term is negligibly small for a sufficiently large $N_A > c$ because

$$\sum_i q_{A,i}E_{B,i} \leq \frac{\kappa_A(c)}{N_A}\sum_i \frac{\eta_i q_{B,i}^2}{1-\gamma_B} = \frac{\kappa_A(c)}{N_A}\kappa_B. \tag{S.92}$$

Note that we have $1-\gamma = \kappa \sum_{i=0} \eta_i/(\kappa + \eta_i N)^2$. The third term is

$$\sum_i \frac{q_{A,i}^2}{1-\gamma_A}E_{B,i} \leq \frac{E_A}{1-\gamma_B} = \frac{1}{1-\gamma_B}\frac{\gamma_A}{1-\gamma_A}\frac{\kappa_A^2}{N_A}, \tag{S.93}$$

where we used $q_{B,i} \leq 1$. It decreases to zero as $N_A$ increases. Thus, we have $E_{A \to B}(\rho) \sim 2(1-\rho)E_B$.

#### A.3.2 Derivation of equation 19

$$\frac{E_{A \to B}(1)}{E_B} = \frac{1}{1-\gamma_A}\frac{\sum_i q_{A,i}^2 q_{B,i}^2 \eta_i^2}{\sum_i q_{B,i}^2 \eta_i^2} \tag{S.94}$$

$$\geq \frac{1}{1-\gamma_A}\left(\frac{\kappa_A}{\kappa_A + \sum_i q_{B,i}^2 \eta_i^3 N_A}\right)^2 \tag{S.95}$$

where the second line comes from Jensen's inequality. Using (S.90), we have

$$\sum_i q_{B,i}^2 \eta_i^3 \leq \frac{\kappa_B(c)^2}{N_B^2}\sum_i \eta_i, \tag{S.96}$$

for $N_B \geq c$. Since we assumed the finite trance of NTK (i.e., $\sum_i \eta_i < \infty$), the left-hand side of (S.96) is finite and converges to zero for a sufficiently large $N_B$. Therefore, we have

$$\frac{E_{A \to B}(1)}{E_B} \gtrsim \frac{1}{1-\gamma_A}. \tag{S.97}$$

In contrast, $E_{A \to B}/E_B \leq 1/(1-\gamma_A)$ since $q_A \leq 1$. Thus, the ratio is sandwiched by $1/(1-\gamma_A)$.

#### A.3.3 Sufficient condition for the negative transfer

Let us denote the $i$-th mode of $E_{A \to B}$ as $E_{A \to B,i}$ and that of $E_B$ as $E_{B,i}$. From $E_{A \to B,i} > E_{B,i}$, we have

$$f(q_i) := 2(1-\rho)(1-q_i) + \frac{q_i^2}{1-\gamma} - 1 > 0, \tag{S.98}$$

where $f(q)$ is a quadratic function and takes the minimum at $q^* = (1-\rho)(1-\gamma)$. The condition $f(q) > 0$ holds if and only if

$$f(q^*) = -(1-\gamma)\rho^2 - 2\gamma\rho + \gamma > 0. \tag{S.99}$$

Solving this, we find $\rho < \sqrt{\gamma}/(1+\sqrt{\gamma})$.

## A.4 Proof of Proposition 3.

For $N_A = N_B$, we have $q_{A,i} = q_{B,i} = q_i$. Then,

$$E_{A \to B}(1) = \frac{1}{(1-\gamma)^2} \sum_i q_i^4 \eta_i^2 \tag{S.100}$$

and the generalization error of single-task training is given by

$$E = \frac{1}{1-\gamma} \sum_i q_i^2 \eta_i^2. \tag{S.101}$$

Model average is obtained in Section D and we have $E_{ave} = (1 - \gamma/2)E < E$ because $0 < \gamma < 1$. Thus, we only need to evaluate the relation between $E_{ave}$ and $E_{A \to B}$:

$$E_{ave} - E_{A \to B} = (1 - \frac{\gamma}{2}) \frac{1}{1-\gamma} \sum_i q_i^2 \eta_i^2 - \frac{1}{(1-\gamma)^2} \sum_i \eta_i^2 q_i^4 \tag{S.102}$$

$$= (1 - \frac{\gamma}{2}) \frac{1}{1-\gamma} \kappa^2 \sum_i a_i^2 - \frac{\kappa^2}{(1-\gamma)^2} \sum_i a_i^2 q_i^2 \tag{S.103}$$

$$= \frac{\kappa^2}{(1-\gamma)^2 N} \underbrace{\left\{ \sum_i N^2 a_i^3 (2 - N a_i) - \frac{1}{2} \gamma^2 (3 - \gamma) \right\}}_{=:F} \tag{S.104}$$

where we defined

$$a_i := \frac{\eta_i}{\kappa + \eta_i N}, \tag{S.105}$$

and used $q_i = 1 - a_i N$ and $\gamma = N \sum_i a_i^2$. We can provide a lower bound of $F$ by using the following cubic function:

$$G(a_i) := N^2 a_i^2 (2 - N a_i).$$

We have $0 \le a_i \le 1/N$ by definition. Let us consider a lower bound of $G$ by using its tangent line at $a_i = t/N$ $(0 \le t \le 1)$, that is, $Nt(4 - 3t)a_i - 2t^2(1 - t)$. Define

$$H(a_i) := G(a_i) - (Nt(4 - 3t)a_i - 2t^2(1 - t)).$$

We have $H(0) = 2t^2(1 - t) \ge 0$. Since $H(1/N) = -2(t - 1)^2(t - 1/2)$, we need $t \le 1/2$ to guarantee $H(a_i) \ge 0$ for all $0 \le a_i \le 1/N$. Here, note that $H$ is a cubic function of $a_i$ and has two fixed points $a_i = t/N$ and $(4 - 3t)/(3N)$. We can see that for $t \le 1/2$, $H$ has the local minimum at $a_i = t/N$ and

$$H(t/N) = t^2(2 - t) - (t^2(4 - 3t) - 2t^2(1 - t))$$
$$= 0. \tag{S.106}$$

Therefore, we have $H(a_i) \ge 0$ for all $0 \le a_i \le 1/N$ when the fixed constant $t$ satisfies $t \le 1/2$. Thus, we have the lower bound of $G$:

$$G(a_i) \ge Nt(4 - 3t)a_i - 2t^2(1 - t). \tag{S.107}$$

Using this lower bound, we have

$$F = \sum_i a_i G(a_i) - \frac{1}{2} \gamma^2 (3 - \gamma)$$

$$\ge \gamma t(4 - 3t) - 2t^2(1 - t) - \frac{1}{2} \gamma^2 (3 - \gamma), \tag{S.108}$$

where we used $\gamma = N \sum_i a_i^2$ and $\sum_i a_i = 1$. By setting $t = \gamma/2$, we obtain

$$F \ge \frac{\gamma^2}{2}(4 - 3\gamma/2) - 2 \frac{\gamma^2}{4}(1 - \gamma/2) - \frac{1}{2} \gamma^2 (3 - \gamma)$$
$$= 0. \tag{S.109}$$

Therefore, we have $E_{ave} > E_{A \to B}$. □

## A.5 Multiple descent

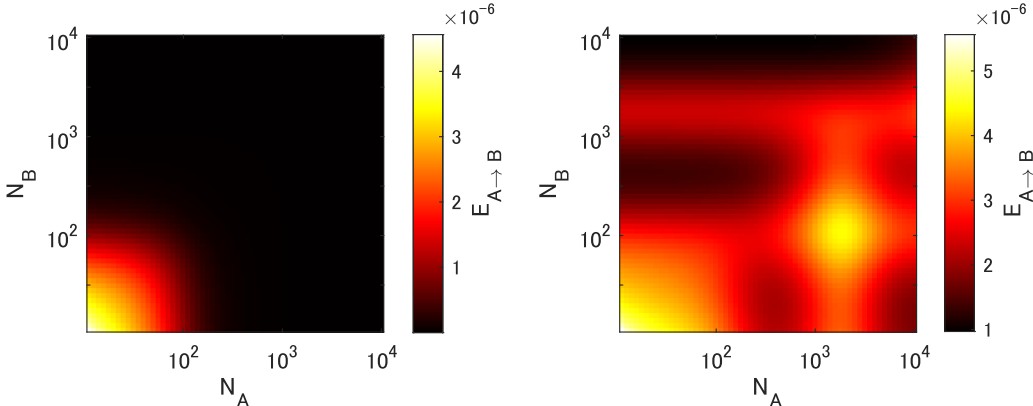

Figure 5: Theoretical values of $E_{A \to B}(1)$. (Left) noise-less case ($\sigma^2 = 0$), (Right) noisy case ($\sigma^2 = 10^{-5}$). We set $D = 100$ and other settings are the same as in Figure 1. In noise-less case, generalization error monotonically decrease as $N_A$ or $N_B$ increases. In contrast, when noise is added, it shows multiple descents.

We overview the multiple descent of $E_1$ investigated in Canatar et al. (2021). It appears for $\sigma > 0$. Let us set $N_B = \alpha D^l$ ($\alpha > 0$, $l \in \mathbb{N}$, $D \gg 1$), which is called the $l$-th learning stage. We have $E_{B,i} = 0$ ($i < l$), $E_{B,l}(\alpha)$ and $\eta_i^2$ ($i > l$). $E_{B,l}(\alpha)$ is a function of $\alpha$, and becomes a one-peak curve depending on the noise and the decay of kernel's eigenvalue spectrum. Because each learning stage has a one-peak curve, the whole learning curve shows multiple descents as the sample size increases. Roughly speaking, the appearance of the multiple descent is controlled by $\gamma$. The $\gamma$ is determined by the kernel's spectrum and sample size $N$. For example, previous studies showed that if we assume the $l$-th learning stage, $\gamma$ is a non-monotonic function of $\alpha$, the maximum of which is $\gamma = 1/(\sqrt{\bar{\lambda}_l} + \sqrt{\bar{\lambda}_l + 1})^2$ for a constant $\bar{\lambda}_l = \sum_{k>l} \bar{\eta}_k/\bar{\eta}_l$,

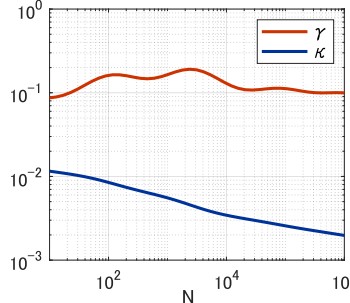

Figure 4: Typical behaviors of $\kappa$ and $\gamma$.

where $\bar{\eta}$ is a normalized eigenvalue of the kernel (Canatar et al., 2021). This tells us that $\gamma$ repeats the increase and decrease depending on the increase of the learning stage as is shown in Figure 4. Roughly speaking, this non-monotonic change of $\gamma$ leads to the multiple descent.

In sequential training, $E_{B,l}(\alpha)$ can similarly cause multiple descent as in single tasks because $E_{A \to B}$ is a weighted summation over $E_{B,i}$. Figure 5 is an example in which noise causes multiple decreases and increases depending on the sample sizes. We set $\rho = 1$ and obtained the theoretical values by (S.22). While $E_{A \to B}(1)$ is a symmetric function of indices A and B for $\sigma = 0$, it is not for $\sigma > 0$. Depending on $N_A$ and $N_B$, the learning "surface" becomes much more complicated than the learning curve of a single task.

A.6 ADDITIONAL EXPERIMENT ON BACKWARD TRANSFER

Figure 6 shows the learning curves of backward transfer. Solid lines show theory, and markers show the mean and interquartile range of NTK regression over 100 trials. The figure shows excellent agreement between theory and experiments. As is shown in Section 4.1, subtle target dissimilarity ($\rho < 1$) causes negative backward transfer (i.e., catastrophic forgetting). For a large $N_A$, $E_{A \to B}^{back}(\rho < 1)$ approaches a non-zero constant while $E_A$ approaches zero. Thus, we have $E_{A \to B}^{back} > E_A$ even for the target similarity close to 1. When sample sizes are unbalanced ($N_A \gg N_B$), the learning curve shows negative transfer even for $\rho = 1$. This is the self-knowledge forgetting revealed in (20). The ratio $E_{A \to B}^{back}(1)/E_A$ takes a constant larger than 1, that is, $1/(1 - \gamma_B)$.

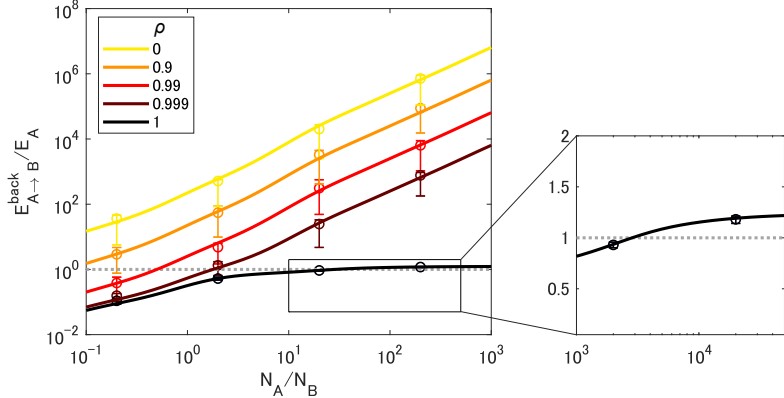

Figure 6: Negative backward transfer easily appears depending on target similarity and sample sizes. We changed $N_A$ and set $N_B = 10^2$. We set $D = 20$ and other experimental details are the same as in Figure 1(b).

## B PROOF OF THEOREM 4.

### B.1 LEARNING CURVE OF MANY TASKS

Eq. (4) is written as

$$f_n(x) = \sum_{k=1}^{n} \Theta(x, X_k)\Theta(X_k)^{-1}(y_k - f_{k-1}(X_k)). \tag{S.110}$$

Each term of this summation is equivalent to the kernel ridge-less regression on input samples $X_k$ and labels $y_k - f_{k-1}(X_k)$. Therefore, we can represent $f_n$ by

$$f_n(x) = \sum_{k=1}^{n} w_k^{*\top}\psi(x) \tag{S.111}$$

with the minimization of the objective function $H$:

$$w_n^* = \lim_{\lambda \to 0} \operatorname{argmin}_{w_n} H(w_n, w_{1:(n-1)}^*), \tag{S.112}$$

$$H(w_n, w_{1:(n-1)}^*) = \frac{1}{2\lambda} \sum_{\mu=1}^{N_n} \left( w_n^\top \psi(x^\mu) - (y_n^\mu - \sum_{k=1}^{n-1} w_k^{*\top}\psi(x^\mu)) \right)^2 + \frac{1}{2}\|w_n\|_2^2 \tag{S.113}$$

$$= \frac{1}{2\lambda} \sum_{\mu=1}^{N_n} \left( (w_n - (\bar{w} - \sum_{k=1}^{n-1} w_k^*))^\top \psi(x^\mu) - \varepsilon_n^\mu \right)^2 + \frac{1}{2}\|w_n\|_2^2. \tag{S.114}$$

The generalization error is

$$E_{n+1} := \left\langle \int dx p(x)(\bar{f}(x) - f_{n+1}(x))^2 \right\rangle_{\mathcal{D}} \tag{S.115}$$

$$= \left\langle \int dx p(x)((w_{n+1}^* - (\bar{w} - \sum_{k=1}^{n} w_k^*))^\top \psi(x))^2 \right\rangle_{\mathcal{D}} \tag{S.116}$$

where $\mathcal{D} = \{\mathcal{D}_1, ..., \mathcal{D}_n\}$.

Since the data samples are independent of each other among different tasks, we can apply Lemma 2 sequentially from $w_{n+1}$ to $w_1$. First, we take the average over the $(n+1)$-th task, that is,

$$E_{n+1|1:n} = \left\langle \int dx p(x)((w_{n+1}^* - (\bar{w} - \sum_{k=1}^{n} w_k^*))^\top \psi(x))^2 \right\rangle_{\mathcal{D}_{n+1}} \tag{S.117}$$

This corresponds to Lemma 2 with $\phi = \eta$ and $u = \bar{w}_A = \bar{w} - \sum_{k=1}^{n} w_k^*$. We have

$$E_{n+1|1:n} = \sum_{i=1}^{n} \phi_{1,i}(\bar{w}_i - \sum_{k=1}^{n} w_{k,i}^*)^2 + R_1 \sigma^2 \tag{S.118}$$

with

$$\phi_{1,i} := \frac{\eta_i q_{n+1,i}^2}{1 - \gamma_{n+1}}, \ R_1 := \frac{\gamma_{n+1}}{1 - \gamma_{n+1}}. \tag{S.119}$$

Here, $\kappa_n, \gamma_n$ and $q_{n,k}$ are determined by $N_n$. Next, we take the average over the $n$-th task:

$$E_{n+1|1:(n-1)} = \left\langle E_{n+1|1:n} \right\rangle_{\mathcal{D}_n, \varepsilon_n} \tag{S.120}$$

$$= \left\langle \sum_i \phi_{1,i}(w_{n,i}^* - (\bar{w}_i - \sum_{k=1}^{n-1} w_{k,i}^*))^2 \right\rangle_{\mathcal{D}_n} + R_1 \sigma^2 \tag{S.121}$$

This corresponds to Lemma 2 with $\phi = \phi_1$ and $u = \bar{w}_A = \bar{w} - \sum_{k=1}^{n-1} w_k^*$. We obtain

$$E_{n+1|1:(n-1)} = \sum_{i=1}^{n} \phi_{2,i}(\bar{w} - \sum_{k=1}^{n-1} w_k^*)^2 + R_2 \sigma^2 \tag{S.122}$$

with

$$\phi_{2,i} = \phi_{1,i} q_{n,i}^2 + \frac{N_n}{\kappa_n^2(1 - \gamma_n)} \eta_i q_{n,i}^2 \sum_j \eta_j \phi_{1,j} q_{n,j}^2, \tag{S.123}$$

$$R_2 = R_1 + \frac{N_n}{\kappa_n^2(1 - \gamma_n)} \sum_j \eta_j \phi_{1,j} q_{n,j}^2. \tag{S.124}$$

Similarly, we can take the averages from the $(n-1)$-th task to the first task, and obtain

$$E_{n+1} = \sum_{i=1}^{n} \phi_{n+1,i} \bar{w}_i^2 + R_{n+1} \sigma^2, \tag{S.125}$$

$$\phi_{m+1,i} = \phi_{m,i} q_{n-m+1,i}^2 + \frac{N_{n-m+1}}{\kappa_{n-m+1}^2(1 - \gamma_{n-m+1})} \eta_i q_{n-m+1,i}^2 \left( \sum_j \eta_j \phi_{m,j} q_{n-m+1,j}^2 \right), \tag{S.126}$$

for $m = 1, ..., n$, and

$$R_{n+1} = \sum_{m=1}^{n} \frac{N_{n-m+1}}{\kappa_{n-m+1}^2(1 - \gamma_{n-m+1})} \sum_j \eta_j \phi_{m,j} q_{n-m+1,j}^2. \tag{S.127}$$

They are general results for any $N_n$. By setting $N_n = N$ for all $n$, we can obtain a simpler formulation. In a vector representation, $\phi_{n+1}$ is explicitly written as

$$\phi_{n+1} = \frac{1}{1 - \gamma} \left[ \text{diag}(q^2) + \frac{N}{(1 - \gamma)\kappa^2} \tilde{q}\tilde{q}^\top \right]^n \tilde{q}. \tag{S.128}$$

Finally, by taking the average over $\bar{w}_i \sim \mathcal{N}(0, \eta_i)$ which is the one-dimensional case of (S.20 ), we have

$$E_{n+1} = \frac{1}{(1 - \gamma)^2} \tilde{q}^\top \left[ \text{diag}(q^2) + \frac{N}{(1 - \gamma)\kappa^2} \tilde{q}\tilde{q}^\top \right]^{n-1} \tilde{q} + R_{n+1} \sigma^2 \tag{S.129}$$

with

$$R_{n+1} = \frac{N}{\kappa^2(1 - \gamma)^2} \sum_{m=1}^{n} \tilde{q}^\top \left[ \text{diag}(q^2) + \frac{N}{(1 - \gamma)\kappa^2} \tilde{q}\tilde{q}^\top \right]^{m-1} \tilde{q} + \frac{\gamma}{1 - \gamma}. \tag{S.130}$$

## B.2 Monotonic decrease of $E_n$

For $\sigma^2 = 0$, we have

$$E_{n+1} = \frac{1}{(1-\gamma)^2} \tilde{q}^\top \mathcal{Q}^{n-1} \tilde{q}. \tag{S.131}$$

Note that $\mathcal{Q}$ is a positive semi-definite matrix. If the maximum eigenvalue is no greater than 1, $E_n$ is monotonically non-increasing with $n$. We denote the infinite norm as $\|A\|_\infty = \max_i \sum_j |A_{ij}|$. Because the infinite norm bounds the eigenvalues, we have

$$\lambda_1(\mathcal{Q}) \le \max_i \sum_j |\mathcal{Q}_{ij}| \tag{S.132}$$

$$= \max_i \underbrace{\left(1 + \frac{N}{\kappa}\eta_i\right)\left(\frac{\kappa}{\kappa + \eta_i N}\right)^2}_{:=G(\eta_i)}, \tag{S.133}$$

where $G(\eta)$ is monotonically decreasing for $\eta \ge 0$ and $G(0) = 1$. Therefore, the largest eigenvalue $\lambda_1(\mathcal{Q})$ is upper-bounded by 1, and $E_n$ becomes monotonically decreasing. In particular. if $\eta_i > 0$ for all $i$, we have $\lambda_1(\mathcal{Q}) < 1$ and obtain $E_{n+1} < E_n$.

For $\lambda_1(\mathcal{Q}) < 1$, it is also easy to check that the series (S.130 ) in $R_{n+1}$ converges to a constant for large $n$. $\qquad\square$

## C Derivation of KRR-like expression

KRR-like expression comes from the inverse formula of a block triangular matrix. Let us write (23) as

$$f_n(x') = [\Theta(x', 1 : (n-1))\ \Theta(x', n)] Z_n^{-1} \begin{bmatrix} y_{1:(n-1)} \\ y_n \end{bmatrix}, \tag{S.134}$$

where $Z_n$ is a block triangular matrix recursively defined by

$$Z_n = \begin{bmatrix} Z_{n-1} & O \\ \Theta(n, 1 : (n-1)) & \Theta(n) \end{bmatrix}. \tag{S.135}$$

Then, (S.134 ) is transformed to

$$[\Theta(x', 1 : (n-1))\ \Theta(x', n)] \begin{bmatrix} Z_{n-1}^{-1} & O \\ -\Theta(n)^{-1}\Theta(n, 1 : (n-1))Z_{n-1}^{-1} & \Theta(n)^{-1} \end{bmatrix} \begin{bmatrix} y_{1:(n-1)} \\ y_n \end{bmatrix} \tag{S.136}$$

$$= \Theta(x', 1 : (n-1))Z_{n-1}^{-1}y_{1:(n-1)} + \Theta(x', n)\Theta(n)^{-1}(y_n - \Theta(n, 1 : (n-1))Z_{n-1}^{-1}y_{1:(n-1)}) \tag{S.137}$$

$$= f_{n-1}(x') + \Theta(x', n)\Theta(n)^{-1}(y_n - f_{n-1}(X_n)). \tag{S.138}$$

Thus, one can see that the KRR-like expression is equivalent to our continually trained model.

## D Derivation of model average

For clarity, we set $\sigma = 0$, $N_A = N_B$, and $\rho = 1$ (that is, $\bar{w}_A = \bar{w}_B$). Generalization error of a model average is expressed by

$$E_{ave} = \left\langle \int dx p(x) \left( \bar{f}_B(x) - \frac{f_A(x) + f_B(x)}{2} \right)^2 \right\rangle_{\mathcal{D}} \tag{S.139}$$

$$= \left\langle \int dx p(x) \left( \left( \bar{w}_B - \frac{w_A + w_B}{2} \right)^\top \psi(x) \right)^2 \right\rangle_{\mathcal{D}} \tag{S.140}$$

$$= \left\langle \left( \bar{w}_B - \frac{w_A + w_B}{2} \right)^\top \Lambda \left( \bar{w}_B - \frac{w_A + w_B}{2} \right) \right\rangle_{\mathcal{D}} \tag{S.141}$$

We can evaluate $E_{ave}$ in a similar way to $E_{A \to B}$. First, take the average over $\mathcal{D}_B$. We define

$$E_{ave|A} = \frac{1}{4} \left\langle ((w_B - (2\bar{w}_B - w_A))^\top \Lambda(w_B - (2\bar{w}_B - w_A)) \right\rangle_{\mathcal{D}_B}. \tag{S.142}$$

This average corresponds to Lemma 2 with replacement from the target task A to B, $\phi \leftarrow \eta$ and $u \leftarrow 2\bar{w}_B - w_A$. Then, we have

$$E_{ave|A} = \frac{1}{4} \sum_{i=0} \left\{ (w_{A,i} - \bar{w}_{B,i})^2 - 2\bar{w}_{B,i}(w_{A,i} - \bar{w}_{B,i})q_{B,i} + \left( \bar{w}_{B,i}^2 + \frac{\eta_i N_B}{\kappa_B^2} E_B \right) q_{B,i}^2 \right\} \eta_i \tag{S.143}$$

$$= \frac{1}{4} \sum_{i=0} (w_{A,i} - (1 + q_{B,i})\bar{w}_{B,i})^2 \eta_i + \frac{\gamma_B}{4} E_B. \tag{S.144}$$

Next, we take the average over $\mathcal{D}_A$. The first term of (S.144) corresponds to Lemma 2 with replacement $\phi \leftarrow \eta$ and $u \leftarrow (1 + q_{B,i})\bar{w}_B$. Then, we get

$$E_{ave} = \langle E_{ave|A} \rangle_{D_A} \tag{S.145}$$

$$= \frac{1}{4} \sum_{i=0} ((1 - q_{A,i})\bar{w}_{A,i} - (1 + q_{B,i})\bar{w}_{B,i})^2 \eta_i + \frac{\gamma_A}{4} E_A + \frac{\gamma_B}{4} E_B. \tag{S.146}$$

For $N_A = N_B$, we have $q_A = q_B$, $\gamma_A = \gamma_B$, and $E_A = E_B$. In addition, since we focused on $\bar{w}_A = \bar{w}_B$, we have

$$E_{ave} = \sum_{i=0} \eta_i \bar{w}_{B,i}^2 q_{B,i}^2 + \frac{\gamma_B}{2} E_B \tag{S.147}$$

$$= \left( 1 - \frac{\gamma_B}{2} \right) E_B \tag{S.148}$$

$\square$

# E    EXPERIMENTAL SETTINGS

## E.1    DUAL REPRESENTATION

In dual representation, the targets (10) are given by

$$[\alpha_{A,i}, \alpha_{B,i}] \sim \mathcal{N}(0, \begin{bmatrix} 1 & \rho \\ \rho & 1 \end{bmatrix} / P'), \tag{S.149}$$

$$\bar{f}_A(x) = \sum_j^{P'} \alpha_{A,j} \Theta(x'_j, x), \ \bar{f}_B(x) = \sum_j^{P'} \alpha_{B,j} \Theta(x'_j, x) \tag{S.150}$$

for a sufficiently large $P'$. We sampled $x'$ from the same input distribution $p(x)$ and set $P' = 10^4$ in all experiments. It is easy to check that targets (10) and (S.150) are asymptotically equivalent for a large $P'$. Because $\Theta(x', x) = \sum_i \psi_i(x')\psi_i(x)$, we have $\sum_j^{P'} \alpha_{A,j} \Theta(x'_j, x) = \sum_i \sum_j^{P'} (\alpha_{A,j} \psi_i(x'_j))\psi_i(x) \sim \sum_i \bar{w}_i \psi_i(x)$, where $\bar{w}_i$ is sampled from $\mathcal{N}(0, \eta_i)$.

## E.2    SUMMARIES ON EXPERIMENTAL SETUP

**Figure 1(a).** We trained the deep neural network (1) with ReLU, $L = 3$, and $M_l = 4,000$ by the gradient descent. We set a learning rate 0.5 and trained the network during 10,000 steps over 50 trials with different random seeds. The target is given by (S.150) with $C = 1$, $D = 10$, and Gaussian input samples $x_i \sim \mathcal{N}(0, 1)$. We initialized the network with $(\sigma_w^2, \sigma_b^2) = (2, 0)$, set $N_A = N_B = 100$, and calculated the generalization error over $4,000$ samples. Each marker shows the mean and interquartile range over the trials. To calculate the theoretical curves, we need eigenvalues $\eta_i$ of the NTK. We numerically compute them by the Gauss-Gegenbauer quadrature following the instruction of Bordelon et al. (2020). Its implementation is also given by Canatar et al. (2021). Since the input

samples are defined on a hyper-sphere $\mathbb{S}^{d-1}$ and the NTK is a dot product kernel (i.e., $\Theta(x', x)$ can be represented by $\Theta(x'^\top x)$), the NTK can be decomposed into Gegenbauer polynomials:

$$\Theta(z) = \sum_{i=0}^{\infty} \eta_i N(d, i) Q_i(z), \tag{S.151}$$

where $\{Q_i\}$ are the Gegenbauer polynomials and $N(d, i)$ are constants depending $d$ and $i$. Since the Gegenbauer polynomials are orthogonal polynomials, we have

$$\eta_i = c \int_{-1}^{1} \Theta(z) Q_i(z) d\tau(z), \tag{S.152}$$

for a certain constant $c$ and $d\tau(z) = (1 - z^2)^{(d-3)/2} dz$. The Gauss-Gegenbaur quadrature approximates this integral by

$$\eta_i \approx c \sum_{j=1}^{r} w_j \Theta(z_j) Q_i(z_j), \tag{S.153}$$

where $w_j$ are constant weights and $z_j$ are the $r$ roots of $Q_r(z)$. The definitions of constants ($N(d, i)$, $c$ and $w_j$) are given in Section 8 of Bordelon et al. (2020) and we set $r = 1000$ as is the same in this previous work. We computed $\eta_i$ ($i = 1, .., 1000$) and substituted them to the analytical expressions.

**Figure 1(b).** This figure shows the results of NTK regression; (2) for the single task and (4) for sequential training. We set $N_B = 4, 000$, $D = 50$ and other settings were the same as in Figure 1(a).

**Figure 2.** This figure shows the results of NTK regression; we set $D = 20$, and other settings are the same as Figure 1(a).

**Figure 3. (NTK regression)** This figure shows the results of NTK regression; We set $D = 10$ and show the means and error bars over 100 trials. Other settings were the same as in Figure 1(a).

**(MLP)** We trained the deep neural network (1) with ReLU, $L = 5$, and $M_l = 512$ by SGD with a learning rate 0.001, momentum 0.9, and mini-batch size 32 over 10 trials with Pytorch seeds 1-10. We set the number of epochs to 150 and scaled the learning rate $\times 1/10$ at the 100-th epoch, confirming that the training accuracy reached 1 for each task.

**(ResNet-18)** We trained ResNet-18 by SGD with a learning rate 0.01, momentum 0.9, and mini-batch size 128 over 10 trials with Pytorch seeds 1-10. We set the number of epochs to 150 and scaled the learning rate $\times 1/10$ at the 100-th epoch, confirming that the training accuracy reached 1 for each task.

Note that the purpose of these experiments was not to achieve performance comparable to state-of-the-art but to confirm self-knowledge transfer and forgetting. Therefore, we did not apply any data augmentation or regularization method such as weight decay and dropout.

