# OpenReview forum: "Learning curves for continual learning in neural networks: Self-knowledge transfer and forgetting"
_ICLR.cc/2022/Conference — ICLR 2022 Poster_

### Official Review · Reviewer_ZHBA · 2021-11-01

**Correctness:** 4
**Technical Novelty And Significance:** 3
**Empirical Novelty And Significance:** Not applicable
**Recommendation:** 8
**Confidence:** 2

**Main Review:**

This is a very well written paper. I never learned the replica trick and I'm not an expert of NTK theory. But the authors provide just enough background detail on the NTK and the base Bordelon et al. (2020) and Canatar et al. (2021) papers to have a grasp of the main tools used to derive the new results. These new results are concisely presented and appropriately discussed. Overall, I found the paper a pleasure to read.

The theory is approximate: it is developed for infinitely-wide networks, and it is not exact even for such networks, as it rests upon some "tricks" that are part of the replica method. The authors therefore carefully validate their semi-analytical results with actual neural network simulations. The theory is in impressive agreement with the large-network simulations. The fit of Fig. 1b, which shows rich nonlinear behavior, is particularly remarkable.

The paper contains a number of interesting results. It justifies the adjective "catastrophic" in "catastrophic forgetting" and it discovers complex transfer behavior as the relative training set size and teacher noise levels are varied. I think that the simulations alone are worth publishing; the fact that they can be described with semi-analytical methods is a strong plus.

A weakness of the paper, which limits the generality of its conclusions, is its focus on the restricted case where p(x) is assumed to be the same for every task.

Like other studies in the NTK regime, another obvious weakness of the paper is the lack of experiments that try to validate to which extent the theory holds when the network width becomes smaller.

I was not capable of reviewing the calculations presented in the appendix. Therefore I cannot stand for the correctness of the results presented in the paper. As I am not an expert in this field, it is also possible that I missed some important related work that should have been cited.

Some other small remarks:
- It would be useful to provide some more detail as to how the $\eta_i$ are computed.
- There are typos and small English mistakes everywhere that should be fixed.
- The discussion section would benefit from some comments on which (if any) existing continual learning algorithms that are designed to reduce forgetting might be analyzed in the NTK regime, with similar tools as those used here.


**Summary Of The Paper:**

The paper presents a study of continual learning of neural networks in the neural tangent kernel (NTK) regime. The authors extend recent work which estimates the generalization error of such networks using a statistical mechanics technique known as the replica trick. The theory makes a number of predictions regarding backward and forward transfer learning which describe well the behavior of finite neural networks trained with gradient descent.

**Summary Of The Review:**

This is a well-presented theoretical study that applies previously developed techniques to analytically describe transfer/continual learning in neural networks. The results are to the best of my knowledge novel. The theory reveals interesting rich phenomena that occurs in actual neural network simulations.

---

> ### Author Response · Authors · 2021-11-17
> **Response to Reviewer ZHBA**
>
> Thanks for your constructive suggestions and for acknowledging the interesting points of our work.
> Following your suggestions, we have fixed English typos and added the following descriptions to the revised manuscript. We expect that our work will definitely serve as a foundation to attack more challenging problems (e.g., the shift of input distribution $p(x)$, finite width correction of NTK theory) in future works.
>
> >some more detail as to how the $\eta_i$ are computed.
>
> We have added some details in Section E.2. Gauss-Gegenbauer quadrature computes the eigenvalues by the summation over NTK, Gegenbauer polynomial, and some coefficients.
>
> >existing continual learning algorithms … might be analyzed in the NTK regime
>
> For example, orthogonal projection methods summarized in Doan et al., 2021 modify only the direction of gradients, and would go well with the NTK formulation. Replay methods allow the reuse of training samples in the previous tasks and seem to require some modification of our calculation (that is, we need to take the saddle point of the replica method simultaneously over different tasks because they share the same replayed samples), but such calculations seem relatively straightforward and would be an interesting direction for both theory and practice. We have mentioned these topics in the Discussion section of the revised manuscript.

---

> > ### Comment · Reviewer_ZHBA · 2021-11-29
> > **I keep my vote for acceptance**
> >
> > Thank you for the response.
> > I remain convinced that the paper is an interesting contribution that should be published at ICLR.

---

### Official Review · Reviewer_2847 · 2021-11-01

**Correctness:** 4
**Technical Novelty And Significance:** 2
**Empirical Novelty And Significance:** Not applicable
**Recommendation:** 6
**Confidence:** 3

**Main Review:**


The paper is well written and well structured. Each section is clear in what the authors try to explain demonstrate.
Nevertheless, the vocabulary used is a bit different from usual continual learning papers, which might be a bit misleading.
A head in CL literature refers often to a group of output neurons while it seems that in this paper it refers only to one output neuron.
"Target function" is also unusual to refer to the function specific to a group of targets. This terminology does not hurt but it should be explained earlier in the paper to avoid confusion.

I acknowledge the use of a theoretical approach to continual learning to better define and understand the problem we are solving.
On the other hand, it is a bit difficult to link the paper setting, with continual learning classical settings.
Continual learning is usually about learning in a non-iid setting with several tasks, where each task might lead to forgetting of the previous one.
In this paper, two different tasks are not supposed to interfere with the other as long as they have different target functions "in the NTK regime, the interaction between different heads do not cause knowledge transfer and forgetting".
The concept of "self-knowledge transfer" is not very intuitive, if I understand correctly is about learning again what the model is supposed to already know.
But it does not learn it from himself (i.e. from its own knowledge) but still from the data... that why "self-knowledge transfer" terminology seems to not fit the phenomenon that the authors want to describe. "self-knowledge transfer" would better fit to a pseudo rehearsal method or to a generative replay method.


Questions:
- Section 4.3 : "the training on task B degrades generalization even though both tasks A and B learn the same target."
 How is it possible? is it overfitting of the target function or just that A and B tasks are about two different modes of the same distribution?
 is it a setting similar to domain incremental settings?

- Section 4.2 : "positive transfer is guaranteed under equal sample sizes" I believe sample size is important but if N_A == N_B and all the samples are the same,
 there is no reason for enhancement, i.e. positive transfer. On the other hand, if you have the same number of samples but with a lower variability, you can have a negative backward transfer.
What am I missing here?

**Summary Of The Paper:**

The paper is a theoretical paper about continual learning, that studies 2 tasks settings in the NTK regime.
The paper study in particular transfer from tasks with the same target functions and introduce the concept of "self-knowledge transfer", i.e. transfer from two tasks with the same target function.

**Summary Of The Review:**

To conclude, I like the aim at theoretically approaching continual learning but it seems there is still a consequent gap to make the theoretical setting proposed fit an actual continual learning setting.
I still will grade this paper above the acceptance threshold, because I think the paper can be useful for the community.

---

> ### Author Response · Authors · 2021-11-17
> **Response to Reviewer 2847**
>
> Thanks for your insightful feedbacks and for acknowledging the usefulness of our work for the community.
> >This terminology does not hurt but it should be explained earlier in the paper to avoid confusion.
>
> Following your suggestion, we have added some explanation on our settings of heads and targets to the Introduction section.
>
> >The concept of "self-knowledge transfer" is not very intuitive ... it does not learn it from himself (i.e. from its own knowledge) but still from the data
>
> This point is due to the ambiguity of the definition of "knowledge transfer", and I agree that a careful description is needed to convey precise neuances. We would like to emphasize that the model inherits the information of the previous task (i.e., “knowledge”) via the parameter initialization for the training of the subsequent task. If both tasks are similar (or the same), the subsequent task can make effective use of the knowledge embedded in the network as the initialization and the model can find a better solution with higher generalization performance. In that sense, the model learns not only from the subsequent data, but also from himself. This is also the case in knowledge transfer achieved by usual transfer learning. We have added some explanations to clarify those issues in the revised manuscript.
>
>
> **Answers to questions:**
> **Section 4.3**
> >"the training on task B degrades generalization even though both tasks A and B learn the same target." How is it possible?
>
> Intuitively speaking, it is because the limited number of task B’s training samples causes a higher generalization error. Note that the limited number of training samples causes a high generalization error in single-task training. We generate training samples in an i.i.d. manner, and task B’s training samples are different from task A’s training samples with high probability. Even after we train the model by many training samples of task A, sequential training forces the model to fit the limited training samples of task B, and it leads to high generalization error (that is, high bias and variance in general). Mathematically speaking, we have $E_{A \rightarrow B} (1) = \sum_i q_{A,i}^2 E_{B,i}/(1-\gamma_A)$. $E_{B,i}$ is the generalization error of single-task training on task B, and this term takes a large value for a small $N_B$. We have added the above explanation for better understanding to the revised manuscript.
>
> **Section 4.2**
> >all the samples are the same, there is no reason for enhancement, i.e. positive transfer …
>
> Note that we generate i.i.d. training samples *in each task*. Even for the same target function case, the set of task A’s training samples is not the same as that of task B’s training samples. Therefore, the sequential training can observe more training samples than single-task training, and the positive transfer is rational. This also means that negative backward transfer does not appear in the setting of Section 4.2 (i.e., $N_A=N_B$, $\rho=1$).

---

> > ### Comment · Reviewer_2847 · 2021-11-29
> > **I keep my vote**
> >
> > Thanks for the answer to my review. After the clarification made by the authors, I feel the paper deserves a bit more than a 6 recommendation but I am still skeptical about if the theoretical setting proposed fits an actual continual learning problem. Therefore, I do not upgrade my recommendation to 8 and keep it to 6.

---

### Official Review · Reviewer_spQ7 · 2021-11-02

**Correctness:** 4
**Technical Novelty And Significance:** 3
**Empirical Novelty And Significance:** 3
**Recommendation:** 6
**Confidence:** 4

**Main Review:**

Theoretical studies are crucial for understanding how deep neural networks work on the continual learning/sequential learning problem. NTK is one of the powerful theoretical tools for analyzing the behavior of neural networks. This paper extends the studies of NTK, specifically the Spectrum Dependent Learning Curves [1], on the continual learning setting via studying the generalization between the sequential tasks. The analyses reveal the relationship between the transfer/forgetting and the dataset for standard neural networks. However, there are still some drawbacks that may limit the significance of the paper:
1. The paper focuses on studies of the behaviors of the standard networks (NTK) trained under standard strategies. Thus the results mainly reflect the influence of the data characteristics and show fewer insights related to the networks and the training strategies. Given the original results in [1], the results in the paper are obvious, further limiting the paper's significance. The results in the paper so far give limited guidance related to model designing, in my opinion.
2. The studies are restricted to continual learning with explicit task boundaries. It is acceptable. But I suggested the authors claim the settings more explicitly at the beginning of the paper/method for clarity.

[1] Bordelon, Blake, Abdulkadir Canatar, and Cengiz Pehlevan. "Spectrum dependent learning curves in kernel regression and wide neural networks." In International Conference on Machine Learning, pp. 1024-1034. PMLR, 2020.

**Summary Of The Paper:**

This paper studies the generalization/transfer behavior of the continual learning/sequential learning under the neural tangent kernel (NTK) regime. It gives the formulation of the generalization error between two tasks in forwarding and backward ways and analyzes the influences of target similarity and sample sizes. Then the self-knowledge transfer case with the same target function is studied, which shows the universality of the catastrophic forgetting and the influence of the sample size. The results are then generalized to many tasks.

**Summary Of The Review:**

Although the paper's significance is a little limited by the relationship with previous work such as [1] and the restricted scenario, the results (especially the self-knowledge transfer case) are interesting. It could be a starting point for more general and informative analyses of deep continual learning.

---

> ### Author Response · Authors · 2021-11-17
> **Response to Reviewer spQ7**
>
> Thanks for your constructive feedbacks and for acknowledging the interesting points of our paper. As you say, we expect that our work will be a starting point for more general and informative analyses of deep continual learning.
>
> **Concern 1:**
> > fewer insights related to the networks and the training strategies.
>
> Although we described only the fully connected neural network as an example,
> the NTK regime also holds in various architectures including CNNs and ResNets (Yang & Littwin, 2021). The NTK formulation of continual learning (4,5) and all of our results also hold in these architectures. In that sense, our work gives insight into universal behaviors that are common among any architecture in the NTK regime.
> It is also true that each architecture has a different NTK and its eigenvalue spectrum. By substituting specific eigenvalue spectrum to our theory, we may be able to show non-trivial behavior that is specific to certain architecture.
> We expect that such case studies and extensions to other training strategies will be the next-step and our work will serve as a foundation for these topics.
>
> >Given the original results in [1], the results in the paper are obvious, further limiting the paper's significance
>
> Lemma 2 is a relatively straightforward extension of the result in [1], but we would like to emphasize that the derivations of Theorems 1 & 4 are unique to our work because we need to decouple the sequence of training to the generalization error conditioned on the previous task. Furthermore, inequality-based evaluations in Proposition 3 and Section A.3 are also unique to our work. It would be an interesting point that we can compare the generalization errors without depending on the detailed spectrum of eigenvalues.
>
> **Concern 2:**
>  > suggested the authors claim the settings more explicitly at the beginning of the paper/method for clarity
>
> Thanks for your suggestion. We have added some explanation on our settings of continual learning to the Introduction section, whose details were also explained in Sections 2 and 3.1.

---

### Official Review · Reviewer_aqmt · 2021-11-03

**Correctness:** 2
**Technical Novelty And Significance:** 3
**Empirical Novelty And Significance:** 2
**Recommendation:** 5
**Confidence:** 3

**Main Review:**

The paper provides some theoretical insights on the forward and backward transfer in sequential training.  My main concern is that I have some doubts regarding its strong assumptions on the data distribution. As stated in the last paragraph on page 4, the input samples of task A and task B are i.i.d. and generated by the same distribution. This assumption cannot be satisfied in most cases of continual learning, especially for class incremental learning the data distribution of each task has significant differences.  In this sense, I think the claims of this work are more about sequential training on a large dataset rather than continual learning.  And the experiments are insufficient to verify the theoretical results in typical continual learning benchmark tasks. The experiments on CIFAR10 are basically sequential training on chunks of a dataset. I'm wondering how much of these theoretical results can be conveyed to typical continual learning tasks. As in most benchmarks, the sample size of each task is the same. Even when the target function is the same, such as in domain incremental learning, the forgetting would like to happen (e.g. permuted or rotated MNIST tasks). But in Sec.4.3, the authors claim no forgetting appears for equal sample sizes, which is not the case for most continual learning tasks in my experience.

Some detailed issues:
1. Fig.2 does not match the description as the blue line E_A is always larger in the figure.
2. What's the error in Fig.3? The generalization error over the last task?
3. Most experiments results are about forward transfer, I think it would be better to show backward transfer as well.


**Summary Of The Paper:**

This paper provides a theoretical analysis for sequential training based on related NTK results.  It shows the similarity between the target functions is a key factor for forward and backward transfer, and when the target functions are the same the samples size of each task is a key factor to forward and backward transfer. It also shows even a slight difference between the target functions can cause catastrophic forgetting and the forgetting might still happen for the same target function when the sample size of a later task increases.

**Summary Of The Review:**

The theoretical results provided in this paper are kind of interesting but I think they are more about sequential training rather than continual learning. The experiments part is a bit weak to support the claims.
As I'm not familiar with the NTK related work, I didn't check through the proofs of those theorems.

---

> ### Author Response · Authors · 2021-11-17
> **Response to Reviewer aqmt**
>
> We would like to thank the reviewer for the valuable and helpful feedbacks.
>
> **Strong assumptions on the data distribution:**
> > the input samples of task A and task B are i.i.d. and generated by the same distribution
>
>  We agree that investigating the non-i.i.d. case or input distribution shift will be an important direction. However, assuming the i.i.d. inputs generated from the same input distribution is common in the theory of continual learning or transfer learning and seems rational as a first step. For example,
>
> S. Lee, S. Goldt & A. Saxe. Continual learning in the teacher-student setup: Impact of task similarity. ICML 2021
>
> analyzed continual learning of shallow networks and assumed that i.i.d. inputs of tasks A and B obey the same distribution. The following theory on transfer learning also assumed it:
>
> N. Tripuraneni, M. Jordan & C. Jin, On the Theory of Transfer Learning: The Importance of Task Diversity, NeurIPS 2020.
>
> The detailed design of the generating rule of targets $p(y|x)$ is different among papers, but they all assumed the same input distribution $p(x)$ between different targets. Certainly, the shift of input distributions is essential for some benchmarks or applications, but these theoretical works (including ours) tell us that even the shift between two targets ($p_A(y|x)$ and $p_B(y|x)$) alone shows non-trivial and rich behaviors of generalization. Such a target shift will often appear and be also important in some applications (e.g., degradation of output signals over time).
> We have added some explanation on this scope of existing theories in the Discussion section. We expect that our work will encourage many researchers to openly discuss and explore the extension to learning with the input shift (e.g., domain incremental learning) in follow-up works.
>
> > this work are more about sequential training on a large dataset rather than continual learning ...
>
> We would like to emphasize that the purpose of our work is not to create a novel theoretical model to reproduce the behavior of a specific benchmark task, but to investigate the NTK model of continual learning (4) from a broad perspective. As we explained in Section 2 and in Remark of Section 3.1, we allow the revisit to the previous classes (targets) in subsequent tasks and “reveal that when the subsequent task revisits previous classes (targets), generalization performance shows interesting increase and decrease”. This is in contrast to the typical benchmarks you mentioned, which do not allow the revisit and focus on the most challenging situation. However, there will be no necessity to forbid obtaining training samples from the same classes (targets) in practical applications. We think that one fundamental aspect of continual learning will be to train the same model continually under the limitation of sample sizes caused by memory constraints or privacy reasons. In this sense, even the training on chunks of a dataset could be a potential problem of continual learning.
> To avoid confusion, we have added some explanation of our settings mentioned in Sections 2 and 3 to the Introduction section of the revised manuscript.
>
>
> **Detailed issues:**
> The following explanation will clear up your confusion.
> >Fig.2 does not match the description as the blue line $E_A$ is always larger in the figure.
>
> The dashed line in Fig. 2 shows the point where the blue line $E_A$ becomes smaller than yellow one. Thus, you can see that the right side of this dashed line matches the description. Note that the difference between blue and yellow lines is seemingly small for large $N_A$ because we took a log-scale. You can also confirm the significant difference in the enlarged part of our new Fig. 6.
>
> >Most experiments results are about forward transfer, I think it would be better to show backward transfer as well.
>
> As we remarked in Section 4.3, for $\rho=1$, the backward transfer is equivalent to the forward transfer *by definition*. Therefore, most experiments (precisely speaking, Figures 2 and 3) also mean the backward transfer. Regarding $\rho<1$, we have created a new figure (Figure 6 in Section A.6) which is a backward transfer version of Figure 1(b). This figure clarifies that our theory explains well the catastrophic forgetting and self-knowledge forgetting.
>
> >What's the error in Fig.3? The generalization error over the last task?
>
> As we remarked in reply to the above comment, $\rho=1$ is the case where all tasks have the same target function. Error means the generalization error over the target function, and this, of course,
> equals the generalization error over the last task. In more detail, Fig.3(a) is the result of NTK regression and Error means the generalization error measured by the mean squared loss, as is similar to other figures. Error in Fig.3(b,c) means "1 - (Test accuracy [%])/100" as we described in Section 5.1.
>
> Finally, we appreciate your helpful comments again and hope that our answers will resolve your concern and confusion.

---

> > ### Comment · Reviewer_aqmt · 2021-11-23
> > **Feedback to author response**
> >
> > I would suggest the authors replace 'continual learning' with 'sequential training' or 'online learning' in the title and content. The main difference between continual learning and online learning is the assumption of the data distribution. Using the term 'continual learning' here may mislead the audience and the main idea of the paper should be precisely expressed. Even if sequential training can be understood as a special case in continual learning, I don't see it is appropriate using a small part of cases to represent the whole topic.

---

> > > ### Author Response · Authors · 2021-11-25
> > > **Response to reviewer feedback**
> > >
> > > Thank you for your additional suggestion. Following your advice, we will change the title to "Learning curves for sequential training in neural networks: Self-knowledge transfer and forgetting".  We will also replace misleading "continual learning" terms in the content with "sequential training."  Although the revision period has ended on Nov. 22 and we currently cannot modify the manuscript, we will modify it for sure if the paper is accepted.

---

### Author Response · Authors · 2021-11-23
**We thank all the reviewers again for their valuable feedbacks.**

All of the feedbacks were very helpful for us to improve our manuscript further. We have revised our paper accordingly to the reviewers’ suggestions, and they are marked in blue for clarity. Although Reviewer aqmt [R1] seems to be concerned about the limitation of input distribution settings, the other reviewers are also aware of this point and then have judged the accept side recommendation. The current work "reveals interesting rich phenomena that occurs in actual neural network simulations" [R4].  We also expect that it will serve as ”a starting point for more general and informative analyses of deep continual learning” [R2] and be "useful for the community" [R3].

---

### Decision · Program_Chairs · 2022-01-20

**Decision:**

Accept (Poster)

**Comment:**

After carefully reading the reviews and rebuttal, I believe this work is of sufficient quality for acceptance. Understanding continual learning from a theoretical stand point is a very important topic. I find that one of the main issue raised by reviewers was about the exact meaning of Continual learning, and whether what the authors studied was more akin to sequential learning. While I don't mind the term sequential learning, and is quite descriptive of the work as well, I disagree that the considered setup is not continual learning.